# Atmospheric fate of organosulfates through gas-phase and aqueous-phase reactions with hydroxyl radicals: implications for inorganic sulfate formation

**Narcisse Tsona Tchinda[1], Xiaofan Lv[1], Stanley Numbonui Tasheh[2], Julius Numbonui Ghogomu[2,3], and Lin Du[1]**

[1]Qingdao Key Laboratory for Prevention and Control of Atmospheric Pollution in Coastal Cities, Environment Research Institute, Shandong University, Qingdao, 266237, China
[2]Department of Chemistry, Faculty of Science, The University of Bamenda, P.O. Box 39, Bambili–Bamenda, Cameroon
[3]Research Unit of Noxious Chemistry and Environmental Engineering, Department of Chemistry, Faculty of Science, University of Dschang, P.O. Box 67, Dschang, Cameroon

**Correspondence:** Narcisse Tsona Tchinda (tsonatch@sdu.edu.cn) and Lin Du (lindu@sdu.edu.cn)

**Abstract.** Organosulfates are important tracers for aerosol particles, yet their influence on aerosol chemical composition remains poorly understood. This study uses quantum chemical calculations based on density functional theory to explore the reactions of some prevalent organosulfates, specifically methyl sulfate and glycolic acid sulfate, with hydroxyl radicals (HO•) in the gas phase and aqueous phase. Results indicate that all reactions initiate with hydrogen abstraction by HO• from $CH_3$- in methyl sulfate and from $-CH_2$- and -COOH in glycolic acid sulfate, followed by the further reaction of the resulting radicals through self-decomposition, interaction with $O_2$ and, possibly, $O_3$. We found that the hydrogen abstraction from the -COOH group in glycolic acid sulfate could lead to decarboxylation and eventually form similar products as methyl sulfate. The primary reaction products are inorganic sulfate, carbonyl compounds, and formic sulfuric anhydride. Rate constants of $1.14 \times 10^{-13}$ and $6.17 \times 10^{-12}$ $cm^3$ molec.$^{-1}$ s$^{-1}$ at 298.15 K were determined for the gas-phase reactions of methyl sulfate and glycolic acid sulfate, respectively. The former value is consistent with a previous experimental report. Additionally, besides $O_2$ as the primary oxidant in the fragmentation of organosulfates, this study unveils that $O_3$ may be a complementary oxidant in this process, especially in environments enriched with ozone. Overall, this study elucidates mechanisms for HO•-initiated transformation of organosulfates and highlights the potential role of chemical substitution, thereby enhancing our understanding of their atmospheric chemistry and implications for inorganic sulfate formation, which are vital for evaluating their impact on aerosol properties and climate processes.

## 1 Introduction

Atmospheric particulate matter is a complex mixture of inorganic and organic matter, with organic matter typically accounting for 20 %–90 % of the total mass (Hallquist et al., 2009; Stone et al., 2012). The concentration level of particulate matter and its evolution have important impacts on regional air quality, climate change and human health. In this regard, the study of organic aerosol concentration levels, sources, and secondary transformations is key to understanding the formation mechanism of compounds responsible for air pollution. Organosulfates, characterized by sulfate groups ($R-OSO_3H/R-OSO_3^-$, where R is an alkyl or aryl group), constitute the most abundant component of organic aerosols. Organosulfates have been widely detected in aerosol particles from various environments in the Americas, Europe,

Asia, and the Arctic over the past decades (Iinuma et al., 2007; Surratt et al., 2008; Nguyen et al., 2014; Kourtchev et al., 2016; Lin et al., 2014). Due to the presence of -$OSO_3^-$ or -$OSO_3H$ functional groups in their structure, organosulfates are acidic and highly water-soluble, which enables them to enhance the hygroscopicity of aerosols, with potential climate impacts (Chan et al., 2011).

The formation pathways of organosulfates are complex and varied. They have been shown to be generated by non-homogeneous and multiphase reactions (Iinuma et al., 2007; Surratt et al., 2008). For example, Rudziński et al. (2009) and Worton et al. (2011) found that the photo-oxidation of a variety of biogenic volatile organic compounds, such as isoprene, $\alpha$-pinene, and $\beta$-pinene, can lead to the formation of organosulfates. Nozière et al. (2010, 2015) suggested that the reaction products of sulfate anions with isoprene can yield organosulfates. Recent studies uncovered pathways by which organosulfates can be generated from the heterogeneous reaction of $SO_2$ with unsaturated bonds in fatty acids (Shang et al., 2016; Passananti et al., 2016). Organosulfates derived from organic acids were recently identified and characterized in fine particulate matter samples collected in the southeastern US, with glycolic acid sulfate being the most abundant (Hettiyadura et al., 2017).

Despite extensive research on the concentration, composition, and formation mechanisms of organosulfates, the limited knowledge of molecular-level mechanisms of their transformations hinders further understanding of their atmospheric processes as well as their physico-chemical properties (Huang et al., 2015). Organosulfates primarily exist in the particulate phase due to their low volatility (Estillore et al., 2016; George and Abbatt, 2010), although a non-negligible fraction has been shown to always be present in the gas phase (Ehn et al., 2010; Le Breton et al., 2018) where they can react continuously with gas-phase oxidants (e.g., HO• radicals, $O_3$, and $NO_3$ radicals) at or near particle surfaces. The transformation of organosulfates generates not only new organic matter, but also inorganic sulfur species, such as $HSO_4^-$ and $SO_4^{2-}$. Consequently, the conversion of organosulfates can significantly alter the composition and physico-chemical properties of atmospheric particulate matter (Hettiyadura et al., 2017). However, little is known in this regard. A few recorded studies include the chemical transformation of methyl sulfate, ethyl sulfate, and an $\alpha$-pinene-derived organosulfate by heterogeneous HO• oxidation (Kwong et al., 2018; Xu et al., 2022). Based on their observed reaction products, the authors suggested a mechanism proceeding through the formation of an alkoxy radical intermediate followed by fragmentation, yet the mechanism remains not fully elucidated. Moreover, organosulfates that can allow hydrogen abstraction at the $\beta$-position to the sulfate group could give rise to more complex mechanisms driven by the Russell mechanism (Russell, 1957) and the Bennett and Summers reaction (Bennett and Summers, 1974). While the presence of functional groups can exhibit specific fea-

tures in the chemical transformation of organosulfates due to their complexity, their potential impact has not yet been thoroughly investigated.

Glycolic acid sulfate and methyl sulfate are two low molecular weight organic sulfates commonly found in the atmosphere, differing structurally by the $\alpha$-substitution of a hydrogen atom by the carboxyl group. Both organosulfates have been detected at various locations around the world at concentrations in the ranges $1.08 \times 10^6$–$5.01 \times 10^7$ molec. cm$^{-3}$ for methyl sulfate (Hettiyadura et al., 2015; Peng et al., 2021) and $1.16 \times 10^7$–$4.71 \times 10^8$ molec. cm$^{-3}$ for glycolic acid sulfate (Huang et al., 2018; Hettiyadura et al., 2015; Wang et al., 2021; Hughes and Stone, 2019; Cai et al., 2020; Liao et al., 2015). This study employs quantum chemical calculations to explore the transformation pathways of these organosulfates in both gas-phase and aqueous-phase environments, emphasizing the effects of carboxyl substitution. Ultimately, this study assesses the potential for sulfate formation and the atmospheric implications of these reactions.

## 2 Methods

### 2.1 Electronic structure calculations and thermochemistry

Quantum chemical calculations were used to investigate the transformation reactions of two organosulfates (methyl sulfate and glycolic acid sulfate) in the gas phase and aqueous phase. Geometry optimizations of all reaction states on the energy surface were performed with density functional theory using the Gaussian 09 package (Frisch et al., 2013). Pre-optimizations were first carried out using the M06-2X functional (Zhao and Truhlar, 2008) and the 6-31+G(d,p) basis set, and the best structures were further refined by the M06-2X/6-311+g(2df,2pd) method to yield the final structures. Vibrational frequency analysis on the M06-2X/6-311+g(2df,2pd) optimized structures were performed using the same method under the rigid rotor-harmonic oscillator approximation at 298.15 K and 1 atm at the same level of theory to yield zero-point energies and the thermochemical parameters. The continuum solvation model based on the solute electron density was used to model the aqueous phase at the same level of theory as the gas phase. This model is particularly suitable for describing atmospheric processes and effectively resolving the energy barriers (Ostovari et al., 2018; Xu and Coote, 2019; Cheng et al., 2019). In the aqueous phase, the Gibbs free energies are calculated at a standard temperature of 298.15 K and by converting the standard pressure of 1 atm (in the gas phase) to the standard concentration of 1 M. Details are provided in Sect. S1 in the Supplement. Single-point energy calculations on the M06-2X/6-311+g(2df,2pd) structures were performed with the CCSD(T)-F12/cc-pVDZ-F12 method using Orca version 4.2.1 (Riplinger and Neese, 2013). The M06-2X/6-311+g(2df,2pd) method has success-

fully been used in a series of previous studies (Ding et al., 2023; Wang et al., 2024; Cheng et al., 2022) and has been proved to efficiently resolve the transition state configuration. Given the size of the system investigated here, this level of theory appears to be a good compromise between accuracy and computation time.

## 2.2 Reaction kinetics

Reactions were modeled by assuming the pseudo-steady state approximation of the reactant complex formed from the interaction between initial reactants (organosulfate (OS = methyl sulfate or glycolic acid sulfate) and hydroxyl radical (HO•)). Based on this approximation, initial reactants are in equilibrium with the reactant complex that further rearranges through a transition state (TS) configuration to form the product complex according to the following reaction:

$$OS + HO• \leftrightarrow \text{ reactant complex } \rightarrow TS \rightarrow \text{ product complex.}$$
$$\text{(R1)}$$

The reaction kinetics analysis was conducted by applying transition state theory (Truhlar et al., 1996). Based on this theory applied to Reaction (R1), the biomolecular rate constant ($k_{bim}$) is given as

$$k_{bim} = K_{eq}k_{uni}, \tag{1}$$

where $K_{eq}$ is the equilibrium constant of formation of the reactant complex, and $k_{uni}$ is the unimolecular rate constant of the transformation of the reactant complex to the product complex, respectively expressed by the following equations:

$$K_{eq} = \frac{1}{c^0} \times \exp\left(-\frac{\Delta G_{eq}}{RT}\right), \tag{2}$$

$$k_{uni} = \kappa \frac{k_B T}{h} \times \exp\left(-\frac{\Delta G^{\#}}{RT}\right). \tag{3}$$

In the above equations, $\Delta G_{eq}$ is the Gibbs free energy of the formation of the reactant complex from initial reactants, $h$ is Planck's constant, $k_B$ is Boltzmann's constant, $\Delta G^{\#}$ is the Gibbs free energy of the barrier separating the reactant complex from the product complex, $R$ is the molar gas constant, $T$ is the absolute temperature, $c^0$ is the standard concentration (with values of 1 M and $2.46 \times 10^{19}$ molec. cm$^{-3}$ in the aqueous phase and in the gas phase, respectively, at 1 atm and 298.15 K), and $\kappa$ is the Eckart tunneling coefficient (calculated by solving the Schrodinger equation for an asymmetrical one-dimensional Eckart potential (Eckart, 1930)).

While Eq. (1) is valid for calculating the biomolecular rate constants of gas-phase reactions, for aqueous-phase reactions, it is corrected by taking into account the contribution of the molecular diffusion described by Collins–Kimball theory (Collins and Kimball, 1949). This leads to the overall rate constant for Reaction (R1) in the aqueous phase as follows:

$$k_{overall} = \frac{k_{bim} \times k_D}{k_{bim} + k_D}. \tag{4}$$

$k_D$ is the steady-state Smoluchowski rate constant calculated as (Smoluchowski, 1917)

$$k_D = 4\pi R_{OS,HO} D N_A. \tag{5}$$

In Eq. (5), $R_{OS,OH}$ stands for the reaction distance between the organosulfate and hydroxyl radical, defined as the sum of their respective radii, $R_{OS}$ and $R_{OH}$. $N_A$ is the Avogadro number and $D$ is the sum of the diffusion coefficients of reactants (Truhlar, 1985). For a reactant $i$ in water, the diffusion coefficient $D_i$ is given by the Stokes-Einstein approach (Einstein, 1905) as follows:

$$D_i = \frac{k_B}{6\pi \eta R_i}. \tag{6}$$

The radii of reactants (organosulfate and hydroxyl radical), assumed to be spherical, were calculated by Eq. (6) using the Multiwfn software (Lu and Chen, 2012). All numerical values for radii, diffusion coefficients of reactants and the steady-state Smoluchowski rate constants are provided in Tables S1 and S2 in the Supplement.

## 3 Results and discussion

The reactions were explored both in the gas phase and the aqueous phase. A previous study indicated that, in the gas phase, methyl sulfate and glycolic acid sulfate are likely to be hydrated under relevant atmospheric temperature and humidity (Tsona and Du, 2019a). Hence, in this study the reactions with HO• in the gas phase have been explored by considering methyl sulfate hydrates (CH$_3$-O-SO$_3$H•••(H$_2$O)$_{n=0-2}$) and glycolic acid sulfate hydrates (HOOC-CH$_2$-O-SO$_3$H•••(H$_2$O)$_{n=0-2}$).

### 3.1 Mechanism of methyl sulfate reaction with HO• radicals

### 3.1.1 CH$_3$-O-SO$_3$H•••(H$_2$O)$_{n=0-2}$ + HO• reaction in the gas phase

Our calculations indicate that the CH$_3$-O-SO$_3$H•••(H$_2$O)$_{n=0-2}$ reaction with HO• proceeds through the formation of an intermediate reactant complex, HO••••CH$_3$-O-SO$_3$H•••(H$_2$O)$_{n=0-2}$, in which the interaction between HO• and CH$_3$-O-SO$_3$H•••(H$_2$O)$_{n=0-2}$ is mainly established through the hydrogen atom of HO• and one oxygen atom of CH$_3$-O-SO$_3$H. This intermediate reactant complex readily reacts through a hydrogen abstraction from the methyl group of CH$_3$-O-SO$_3$H by HO• to form the H$_2$C•-O-SO$_3$H•••(H$_2$O)$_{n+1}$ product complex. A distinct feature was observed for $n = 2$, where the sulfate group was deprotonated, forming an ion pair with H$_3$O$^+$ (i.e, H$_2$C•-O-SO$_3^-$,H$_3$O$^+$•••(H$_2$O)$_2$). This is likely to be because for this configuration the minimum amount of water (2 molecules) necessary to achieve the stability of H$_3$O$^+$ in

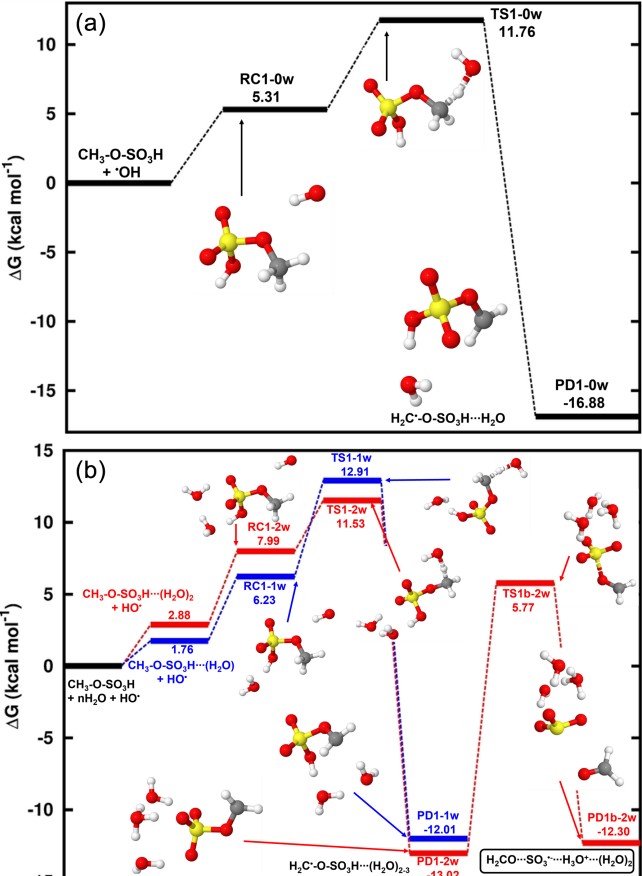

**Figure 1.** Gibbs free energy changes (in kcal mol$^{-1}$) and optimized structures for all intermediates in the reaction of CH$_3$-O-SO$_3$H$\cdots$(H$_2$O)$_{n=0-2}$ with HO• radicals. The top panel is the reaction in the absence of water and the bottom panel is the reaction in the presence of water, where the blue indicates the monohydrated reaction and the red line indicates the dihydrated reaction. The sulfur atom is yellow, the oxygen atom is red, the carbon atom is gray, and the hydrogen atom is white.

the gas phase has been reached (Markovitch and Agmon, 2007; Heindel et al., 2018). Although the presence of water has a weak effect on the formation of the intermediate reactant complex, it substantially stabilizes the transition state towards the formation of the product complex by reducing the Gibbs free energy barrier from 6.45 kcal mol$^{-1}$ for the CH$_3$-O-SO$_3$H + HO• reaction to 3.52 kcal mol$^{-1}$ for the CH$_3$-O-SO$_3$H$\cdots$(H$_2$O)$_2$ + HO• reaction. Energetics and structural details of this reaction are given in Table 1 and Fig. 1. The observed catalytic effect of water has been generally observed in some previous H-abstraction reactions (Wang et al., 2019; Wang et al., 2017; Zhang et al., 2024; Wang et al., 2024).

Unimolecular rate constants for the transformation of HO$\cdots$CH$_3$-O-SO$_3$H$\cdots$(H$_2$O)$_n$ to H$_2$C•-O-SO$_3$H$\cdots$(H$_2$O)$_{n+1}$ at 298 K were determined to be

$1.15 \times 10^8$, $7.89 \times 10^7$ and $1.59 \times 10^{10}$ s$^{-1}$, corresponding to atmospheric lifetimes of 9 ns, 13 ns, and 63 ps, for $n = 0$, 1, and 2, respectively, for the intermediate reactant complex. The lifetimes in the picosecond and nanosecond regimes indicate that HO$\cdots$ • CH$_3$-O-SO$_3$H$\cdots$(H$_2$O)$_{n=0-2}$ complexes are indeed highly reactive and would most likely react fast before they could experience collisions with other abundant atmospheric oxidants (Bork et al., 2012). Hence, the immediate fate of HO$\cdots$ •CH$_3$-O-SO$_3$H$\cdots$(H$_2$O)$_{n=0-2}$ is undoubtedly hydrogen abstraction by HO• to form H$_2$C•-O-SO$_3$H$\cdots$(H$_2$O)$_{n+1}$. The bimolecular rate constant for this reaction is determined to be $1.14 \times 10^{-13}$ cm$^3$ molec.$^{-1}$ s$^{-1}$ at 298.15 K. As a radical, H$_2$C•-O-SO$_3$H$\cdots$(H$_2$O)$_{n+1}$ would eventually undergo further decomposition depending on its stability, atmospheric lifetime, and surrounding atmospheric conditions.

The stability of H$_2$C•-O-SO$_3$H$\cdots$(H$_2$O)$_{n+1}$, examined relative to decomposition through the reverse reaction back to the HO$\cdots$CH$_3$-O-SO$_3$H$\cdots$(H$_2$O)$_n$ complex, reveals barrier heights of 28.64, 24.92, and 24.55 kcal mol$^{-1}$ for $n = 0$, 1, and 2, corresponding to unimolecular rate constants of $6.16 \times 10^{-9}$, $3.30 \times 10^{-6}$, and $6.14 \times 10^{-6}$ s$^{-1}$, respectively. These low-rate constants indicate that, once formed, H$_2$C•-O-SO$_3$H$\cdots$(H$_2$O)$_{n+1}$ would not react back to form initial reactants before possible collision with atmospheric oxidants have occurred. Beside hypothesizing self-decomposition to be a likely outcome of H$_2$C•-O-SO$_3$H, observed atmospheric lifetimes indicate that its fate would depend upon collisions with most atmospheric oxidants, including O$_2$, O$_3$, and NO$_2$. Considering that water is known to evaporate fast from atmospheric species, H$_2$C•-O-SO$_3$H would readily react in both its unhydrated and hydrated forms.

Based on these observations, the fate of H$_2$C•-O-SO$_3$H was examined relative to the following decomposition processes:

$$\text{H}_2\text{C•} - \text{O} - \text{SO}_3\text{H}\cdots(\text{H}_2\text{O})_n \rightarrow \text{H}_2\text{CO}$$
$$+ \text{SO}_3\text{•}^-, \text{H}_3\text{O}^+\cdots(\text{H}_2\text{O})_{n-1}, \quad \text{(R2)}$$

$$\text{H}_2\text{C•} - \text{O} - \text{SO}_3\text{H}\cdots(\text{H}_2\text{O})_n + \text{NO}_2 \rightarrow \text{ON} + \text{H}_2\text{CO}$$
$$+ \text{SO}_4^- \cdots\text{H}_3\text{O}^+\cdots(\text{H}_2\text{O})_{n-1}, \quad \text{(R3)}$$

$$\text{H}_2\text{C•} - \text{O} - \text{SO}_3\text{H}\cdots(\text{H}_2\text{O})_n + \text{O}_3 \rightarrow \text{O}_2 + \text{HCO•}$$
$$+ \text{H}_2\text{SO}_4 + (n-1)\text{H}_2\text{O}, \quad \text{(R4)}$$

$$\text{H}_2\text{C•} - \text{O} - \text{SO}_3\text{H} + \text{O}_2 \rightarrow \rightarrow \text{HC(O)} - \text{O} - \text{SO}_3\text{H}. \quad \text{(R5)}$$

Our attempts to optimize the self-decomposition of H$_2$C•-O-SO$_3$H$\cdots$(H$_2$O)$_n$ failed except for $n = 3$ (Reaction R2). This was as expected since the gas-phase stability of H$_3$O$^+$ can only be achieved by solvation with at least two water molecules (Markovitch and Agmon, 2007; Heindel et al., 2018).

Reaction (R2) proceeds by one S-O bond breaking and a hydrogen transfer from -SO$_3$H to H$_2$O to release the H$_2$CO$\cdots$SO$_3$•$^-$,H$_3$O$^+$$\cdots$(H$_2$O)$_2$ complex. The transition state in this process was located at 18.79 kcal mol$^{-1}$

**Table 1.** Electronic energy changes ($\Delta E$) and Gibbs free energy changes ($\Delta G$ at 298.15 K and 1 atm) for all intermediate species in the reaction of methyl sulfate with HO$^{\bullet}$ radicals. Energy units are kcal mol$^{-1}$. "RC" stands for intermediate reactant complex, "TS" stands for transition state, "PD" stands for product, and "nw" stands for the number of added water molecules to the reaction of methyl sulfate with HO$\bullet$ radicals.

| Reaction | $\Delta G$ | $\Delta E$ |
|---|---|---|
| $CH_3$-O-$SO_3H \bullet\bullet\bullet (H_2O)_{n=0-2} + \bullet OH \leftrightarrow$ RC1-nw $\rightarrow$ TS1-nw $\rightarrow$ PD1-nw ($H_2C\bullet$-O-$SO_3H \bullet\bullet\bullet (H_2O)_{n+1}$) | | |
| $n = 0$ | | |
| RC1-0w | 5.31 | $-2.57$ |
| TS1-0w | 11.76 | 5.66 |
| PD1-0w | $-16.88$ | $-26.17$ |
| $n = 1$ | | |
| RC1-1w | 6.23 | $-13.38$ |
| TS1-1w | 12.91 | $-5.16$ |
| PD1-1w | $-12.01$ | $-32.34$ |
| $n = 2$ | | |
| RC1-2w | 7.99 | $-22.64$ |
| TS1-2w | 11.53 | $-18.10$ |
| PD1-2w | $-13.02$ | $-46.65$ |
| $H_2C\bullet$-O-$SO_3^-$,$H_3O^+ \bullet\bullet\bullet (H_2O)_2 \rightarrow$ TS1b-2w $\rightarrow$ PD1b-2w ($H_2CO \bullet\bullet\bullet SO_3\bullet^-$,$H_3O^+ \bullet\bullet\bullet (H_2O)_2$) | | |
| $H_2C\bullet$-O-$SO_3^-$,$H_3O^+ \bullet\bullet\bullet (H_2O)_2$ | 0 | 0 |
| TS1b-2w | 18.79 | 23.35 |
| PD1b-2w | 0.72 | 7.09 |
| $H_2C\bullet$-O-$SO_3H + O_3 \leftrightarrow$ RC31 $\rightarrow$ TS31 $\rightarrow$ PD31 ($O_2 + \bullet$O-$CH_2$-O-$SO_3H$) $\rightarrow$ RC31b $\rightarrow$ TS31b $\rightarrow$ PD31b | | |
| RC31 | $-18.70$ | $-37.32$ |
| TS31 | $-25.72$ | $-42.17$ |
| PD31 | $-22.48$ | $-34.73$ |
| RC31b | $-4.48$ | 5.22 |
| TS31b | 10.42 | 26.55 |
| PD31b | $-22.24$ | $-7.05$ |
| $H_2C\bullet$-O-$SO_3H + O_2 \rightarrow \bullet$OO-$CH_2$-O-$SO_3H$ | $-45.98$ | $-65.69$ |
| $\bullet$OO-$CH_2$-O-$SO_3H + \bullet$OO-$CH_2$-O-$SO_3H \rightarrow$ RC32 $\rightarrow$ TS32 $\rightarrow$ PD32 | | |
| RC32 | $-99.39$ | 6.28 |
| TS32 | $-83.49$ | 15.33 |
| PD32 | $-234.34$ | $-10.68$ |

above $H_2C\bullet$-O-$SO_3^-$,$H_3O^+ \bullet\bullet\bullet (H_2O)_2$. To confirm the location of the free electron on the different states of $H_2C\bullet$-O-$SO_3H \bullet\bullet\bullet (H_2O)_3$ decomposition, the analysis of the charge distribution was performed. As shown in Fig. 2a, the electronic charge initially located on the $CH_2$- fragment progressively migrates along the $H_2C\bullet$-O-$SO_3H$ core to ultimately rest on $SO_3$, leaving $CH_2O$ electronically neutral. Although $SO_3\bullet^-$ and $H_2CO$ are expected products of this decomposition, our calculations show that the outcome of this reaction can only be moderate due to the relatively high energy barrier. Several experimental and theoretical studies showed that $SO_3^{\bullet-}$ would quickly hydrate in the gas phase to form a $SO_3^{\bullet-} \bullet\bullet\bullet (H_2O)_n$ cluster wherein the oxidation to sulfuric acid occurs in a mechanism stabilized and catalyzed by a free electron (Bork et al., 2013; Tsona and Du, 2019b; Fehsenfeld

and Ferguson, 1974; Svensmark et al., 2007; Enghoff and Svensmark, 2008). Considering this outcome for $SO_3^{\bullet-}$ in the gas phase, $H_2CO$ and inorganic sulfate are expected products of the gas-phase reaction of methyl sulfate with HO$\bullet$ at ambient conditions. The calculated unimolecular rate constant of this decomposition at 298.15 K, $1.03 \times 10^{-1}$ s$^{-1}$, indicates that this process can account for the fate of $H_2C\bullet$-O-$SO_3H$ only under the conditions of low oxidants. The products predicted by our calculations were observed in a previous experimental study by Kwong et al. (2018) for the same reaction, and the mechanisms from Russell (1957) and Bennett and Summers (1974) were speculated by the authors to explain this formation. Besides these mechanisms, Reaction (R2) in this study might be a complementary mechanism to the formation of inorganic sulfate and formaldehyde from the reac-

tion of methyl sulfate with HO•, at least at a certain degree of hydration or humidity.

Besides the self-decomposition of $H_2C\cdot$-O-$SO_3H$, we further examined its reactions with $NO_2$ (Reaction R3), $O_3$ (Reaction R4), and $O_2$ (Reaction R5). We found that for Reaction (R3), $SO_4\cdot^- + H_2CO + NO$ formation would be prevented by a high Gibbs free energy barrier ($\sim$ 74 kcal mol$^{-1}$), regardless of the number of water molecules involved. Energetics of this reaction are given in Table S3 and Fig. S1 in the Supplement. This high energy barrier is likely related to the difficulty in breaking the ON-O and C-O bonds to release the products. We conclude that Reaction (R3) is likely without atmospheric significance to the chemistry of $CH_3$-O-$SO_3H$ under relevant atmospheric conditions.

The reaction of $H_2C\cdot$-O-$SO_3H\cdots(H_2O)_n$ with $O_3$ was examined thereafter. Regardless of the presence or absence of water, this reaction proceeds through a submerged energy barrier, with the general mechanism following two main steps: the interaction of $H_2C\cdot$-O-$SO_3H$ with $O_3$ and the unimolecular decomposition of the resulting intermediate product. The interaction of $H_2C\cdot$-O-$SO_3H\cdots(H_2O)_n$ with $O_3$ led to exergonic formation of the $O_3\cdots H_2C\cdot$-O-$SO_3H\cdots(H_2O)_n$ intermediate reactant complex. The formed reactant complexes are further transformed through an oxygen transfer from $O_3$ to $H_2C\cdot$-O-$SO_3H\cdots(H_2O)_n$ via transition state configurations to form alkoxyl radicals $\cdot$O-$CH_2$-O-$SO_3H\cdots(H_2O)_n$. The energy barrier in this transformation in the absence of water is located 7.02 kcal mol$^{-1}$ below the corresponding reactant complex and is weakly altered by the presence of water. The exergonic formation of $O_3\cdots H_2C\cdot$-O-$SO_3H\cdots(H_2O)_n$ and the submerged barrier in its transformation are indicative of instantaneous formation of $\cdot$O-$CH_2$-O-$SO_3H\cdots(H_2O)_n$ from $H_2C\cdot$-O-$SO_3H\cdots(H_2O)_n$ with $O_3$.

Optimized structures and energetics of all reaction states in these reactions are given in Figs. 3, S2, Tables 1 and S4. The direct product of this transformation is a sulfonate alkoxy radical, $\cdot$O-$H_2C$-O-$SO_3H$, which further decomposes through the C-O bond cleavage and hydrogen transfer from $\cdot$O-$CH_2$- to -$SO_3H$ to form the $H_2SO_4\cdots HCO\cdot$ product complex with substantial energy gain. The electronic charge distribution on $\cdot$O-$H_2C$-O-$SO_3H$ in the reactant and on $HCO^\bullet$ in the product complex was confirmed by our electronic charge analysis as shown in Figs. 2b and S3. The barrier height to this decomposition is significantly lowered from 14.90 kcal mol$^{-1}$ (for $H_2C\cdot$-O-$SO_3H$) to 6.59 kcal mol$^{-1}$ (for $H_2C\cdot$-O-$SO_3H\cdots(H_2O)_3$). These correspond, respectively, to unimolecular rate constants of $7.38 \times 10^1$ and $9.06 \times 10^7$ s$^{-1}$ at 298.15 K. The particular stability of the transition state in $H_2C\cdot$-O-$SO_3H\cdots(H_2O)_3$ decomposition can be attributed to the mediation of the additional water molecule in the hydrogen transfer from $\cdot$O-$CH_2$- to -$SO_3H$. This is in line with the demonstrated increasing catalytic role of water with increasing number of water molecules in $SO_3$ hydrolysis to sulfuric acid (Larson et al., 2000; Hofmann and Schleyer, 1994; Hofmann-Sievert and Castleman, 1984; Mo-

rokuma and Muguruma, 1994; Loerting and Liedl, 2000). The currently presented mechanism for $H_2SO_4\cdots HCO\cdot$ formation from C-O bond cleavage was suggested by Huang et al. (2018) to explain bisulfate formation from the fragmentation of organosulfates. Compared to $H_2C\cdot$-O-$SO_3H$ self-decomposition and reaction with $NO_2$, the reaction with $O_3$ is the most energetically and kinetically favorable process. Nonetheless, considering all the processes by Reactions (R2)–(R4), it is obvious that the main products of the gas-phase reaction of methyl sulfate with HO• are formaldehyde ($H_2CO$) and sulfuric acid ($H_2SO_4$). This agrees with the experimental observation by Kwong et al. (2018).

Contrary to the self-decomposition of $H_2C\cdot$-O-$SO_3H$ and reactions with $NO_2$ and $O_3$ that are favorable with $H_2C\cdot$-O-$SO_3H$ hydrates, we were unable to fully optimize the $O_2$ reaction with $H_2C\cdot$-O-$SO_3H$ hydrates but with the unhydrated system, instead. This led to exergonic formation of the $\cdot$OO-$CH_2$-O-$SO_3H$ peroxyl radical with 45.98 kcal mol$^{-1}$ Gibbs free energy gain. The chemistry of peroxyl radicals has been the subject of several experimental and theoretical studies. It is widely accepted that the peroxyl radical would predominantly decompose to form alkoxyl radicals or alcohols along with carbonyl compounds through tetroxide intermediates (Russell, 1957). However, although the end products from this decomposition have been verified experimentally, the mechanisms have often been deemed unlikely due to inconsistency between thermodynamic experiments and computational studies (Nangia and Benson, 1980; Zhang et al., 2012; Liang et al., 2011). Moreover, the effort to elucidate alcohols and carbonyl compounds formation from the decomposition of the tetroxide in a previous study could not be achieved due to the impossibility of determining the corresponding transition states (Ghigo et al., 2003), while alkoxyl radical formation was observed to simply correspond to the dissociation of the tetroxide. A more recent theoretical study specifically focusing on the decomposition pathways of the tetroxide intermediate indicated that although substantial uncertainties may exist in their computed energetics, alkoxyl radicals are likely primary products from atmospherically relevant peroxyl radicals (Salo et al., 2022). Following the above reasoning, our calculations indicate that two molecules of $\cdot$OO-$CH_2$-O-$SO_3H$ recombine to form a tetroxide that can further decompose to generate two alkoxyl radicals $\cdot$O-$CH_2$-O-$SO_3H$ clustered to molecular oxygen. Then, the two $\cdot$O-$CH_2$-O-$SO_3H$ radicals quickly interact with $O_2$ to form $H_2O_2$ and formic sulfuric anhydride ($HC(O)$-O-$SO_3H$) (Reaction R5). The latter product has been identified to enhance new particle formation (An, 2024). Energetics and structures of all intermediates in this reaction are given in Fig. 3.

Considering the recombination of two $H_2C\cdot$-O-$SO_3H$ molecules as another likely fate of $H_2C\cdot$-O-$SO_3H$ in the absence of other reaction partners such as $O_2$, $O_3$, or $NO_2$ besides self-decay, our preliminary calculations indicate that this recombination would occur fast and would proceed through C-C bond formation via a barrierless process. How-

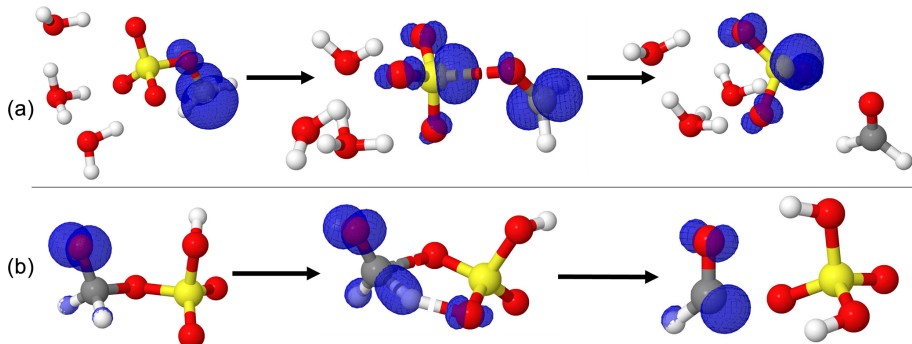

**Figure 2.** Representation of the spin density (blue color) on the electronic states of the decomposition of **(a)** $H_2C\bullet$-O-$SO_3H\bullet\bullet\bullet(H_2O)_3$, and **(b)** $\bullet O$-$H_2C$-O-$SO_3H$. From left to right are the reactant, transition state, and product complex, respectively. The sulfur atom is yellow, the oxygen atom is red, the carbon atom is gray, and the hydrogen atom is white.

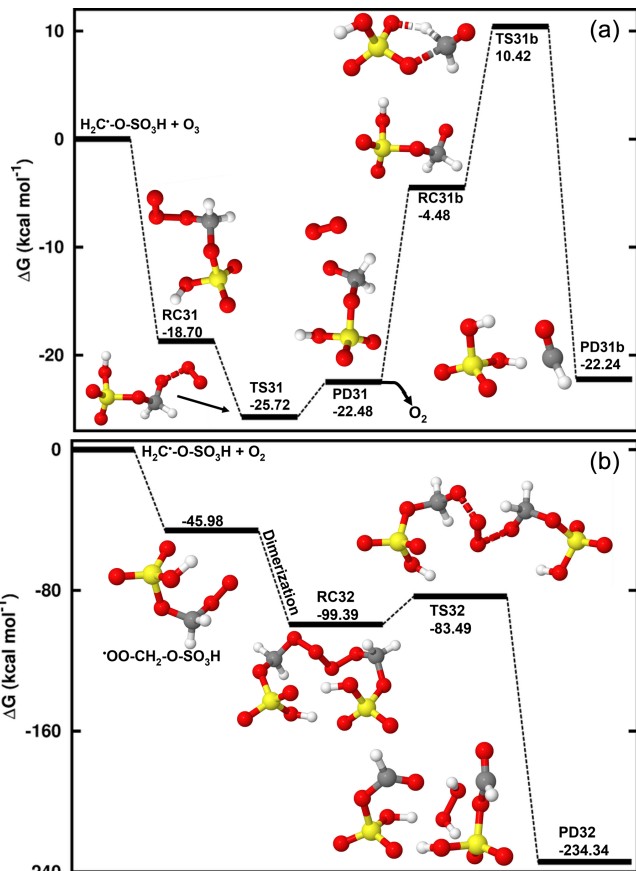

**Figure 3.** Gibbs free energy changes (in kcal mol$^{-1}$) and optimized structures for all intermediates in the reaction of $H_2C\bullet$-O-$SO_3H$ with $O_3$ **(a)** and $O_2$ **(b)**. The sulfur atom is yellow, the oxygen atom is red, the carbon atom is gray, and the hydrogen atom is white.

ever, the relevance of the resulting product is not clearly established and its fate may mostly depend on reactions initiated by a hydrogen abstraction from the -CH$_2$- group.

The studied reaction of methyl sulfate with HO$\bullet$ radicals in the gas phase shows an example of the main processes through which organosulfates may be converted into inorganic sulfates. Humidity is seen to play a non-negligible role in the effective reaction of methyl sulfate while from the kinetics point of view, $O_3$ is a key oxidant besides $O_2$ in the intermediate steps.

### 3.1.2 CH$_3$-O-SO$_3^-$ + HO$\bullet$ reaction in the aqueous phase

In the aqueous phase, methyl sulfate (CH$_3$-O-SO$_3^-$) undergoes similar steps as its electronically neutral counterpart to form the product complex H$_2$C$\bullet$-O-SO$_3^-$ $\bullet\bullet\bullet$H$_2$O. Figure 4 shows the Gibbs free energy surface for this reaction at 298.15 K and 1 M concentration, along with the structures of all stationary states. Further energetics details for this reaction are given in Table 2. We determined a Gibbs free energy barrier of 3.54 kcal mol$^{-1}$ for this reaction (4.69 kcal mol$^{-1}$ lower than in the CH$_3$-O-SO$_3$H + HO$\bullet$ reaction), and a bimolecular rate constant of $7.87 \times 10^6$ M$^{-1}$ s$^{-1}$. This value is about 13 times lower than a previous experimental value (Gweme and Styler, 2024), and the difference can be attributed both to computational errors and environmental factors such as pH, temperature, and ionic strength. The formation of H$_2$C$\bullet$-O-SO$_3^-$ $\bullet\bullet\bullet$H$_2$O occurred with substantial Gibbs free energy gain, $-14.28$ kcal mol$^{-1}$, stable enough toward decomposition to initial reactants for which the energy barrier 23.89 kcal mol$^{-1}$ is found (see Fig. 4). Based on this energy barrier height, we determined an atmospheric lifetime of 0.61 days for H$_2$C$\bullet$-O-SO$_3^-$ $\bullet\bullet\bullet$H$_2$O. This indicates that besides self-decomposition as an alternative chemical fate, H$_2$C$\bullet$-O-SO$_3^-$ $\bullet\bullet\bullet$H$_2$O will live long enough to experience collisions with abundant atmospheric oxidants. While NO$_2$ dissolves effectively in water and is known to react with molecular water to produce nitric acid which is highly soluble in water (Ford and Miranda, 2020; Tan and Piri, 2013; Lee and Schwartz, 1981; England and Corcoran, 1974), the stability of O$_3$ solubility in water is readily affected by various fac-

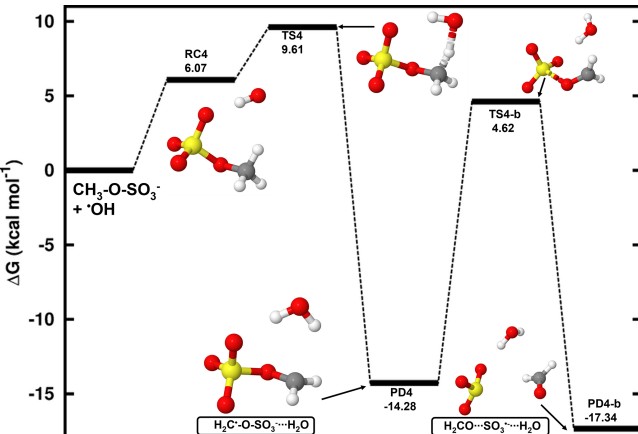

**Figure 4.** Gibbs free energy changes (in kcal mol$^{-1}$) and optimized structures for all intermediates in the reaction of CH$_3$-O-SO$_3^-$ with HO• radicals. The sulfur atom is yellow, the oxygen atom is red, the carbon atom is gray, and the hydrogen atom is white.

tors including ozone concentration, pH, and ultraviolet light (Lovato et al., 2009). Depending on environmental conditions, ozone can react via a direct reaction pathway involving molecular ozone or by an indirect route involving reactive intermediates that arise from its decomposition (Buehler et al., 1984; Buhler et al., 1984; Staehelin et al., 1984; Staehelin and Hoigne, 1982), whereas the reaction with NO$_2$ would be equivalent to explicitly assessing the reaction with NO$_3^-$. Moreover, O$_3$ is known to be 13 times more soluble in water than O$_2$ (Seinfeld and Pandis, 1998). Hence, besides the self-decay reaction, the fate of H$_2$C•-O-SO$_3^-$ was examined via reactions with O$_2$ and O$_3$:

$$H_2C\bullet - O - SO_3^- \rightarrow H_2CO + SO_3\bullet^-, \tag{R6}$$

$$H_2C\bullet - O - SO_3^- + O_3 \rightarrow O_2 + HSO_4^- + HCO\bullet, \tag{R7}$$

$$H_2C\bullet - O - SO_3^- + O_2 \rightarrow \rightarrow HSO_4^- + HCO\bullet. \tag{R8}$$

While inspecting the vibrational modes of H$_2$C•-O-SO$_3^-$, it is obvious that dissociation along the H$_2$C•O-SO$_3$ bond to form H$_2$CO + SO$_3$•$^-$ (Reaction R6) would be a possible chemical fate. The analysis of electronic charge distribution on H$_2$C•-O-SO$_3^-$ confirms that the unpaired electron initially on CH$_2$- gradually migrates to completely rest on SO$_3$ in the products, leaving CH$_2$O uncharged (see Fig. S4). This process was examined and a Gibbs free energy barrier height of 18.90 kcal mol$^{-1}$ was found (see Fig. 4), which corresponds to a unimolecular rate constant of $8.56 \times 10^{-2}$ s$^{-1}$ at 298.15 K. This is nearly equal to the rate constant of the similar step ($1.03 \times 10^{-1}$ s$^{-1}$) in the reaction of CH$_3$-O-SO$_3$H under humid conditions. The predicted relatively low-rate constant of H$_2$C•-O-SO$_3^-$ decomposition to H$_2$CO and SO$_3$•$^-$ can only account for the CH$_3$-O-SO$_3^-$ fate under the conditions of low oxidants concentrations.

Considering the H$_2$C•-O-SO$_3^-$ interaction with O$_3$ (Reaction R7), this reaction is completely downhill, and it follows

two main steps: formation of an alkoxy radical and decomposition of the latter into HSO$_4^-$ and HCO• radicals (see Fig. 5). The first step is highly exergonic, with the reactant complex (O$_3$•••H$_2$C•-O-SO$_3^-$) being formed with $-52.05$ kcal mol$^{-1}$ Gibbs free energy change. The decomposition of O$_3$•••H$_2$C•-O-SO$_3^-$ into the O$_2$••••O-H$_2$C-O-SO$_3^-$ product complex is almost instantaneous, with the Gibbs free energy barrier located 5.98 kcal mol$^{-1}$ below the reactant. As O$_2$ evaporates from the product complex, the resulting alkoxy radical, •O-H$_2$C-O-SO$_3^-$, is rapidly decomposed to HSO$_4^-$ and HCO• by overcoming a relatively low energy barrier (7.92 kcal mol$^{-1}$). The charge analysis confirms that during •O-H$_2$C-O-SO$_3^-$ decomposition, the unpaired electron effectively delocalizes from the oxygen atom of the alkoxy functional group to concentrate on the carbon atom of HCO (see Fig. S4), leaving HSO$_4^-$ as one of the main products. These products were also identified as primary products in the CH$_3$-O-SO$_3^-$ reaction with •OH, although a different formation mechanism was proposed (Kwong et al., 2018). This study demonstrates that the reaction with O$_3$ is distinctly thermodynamically and kinetically favorable, therefore highlighting the presented mechanism to be a determinant step in the oxidation of methyl sulfate by •OH radicals. While the role of HSO$_4^-$ in the atmosphere is clearly established, for example in aerosol formation, HCO• has never been observed in aqueous media. Its presence has only been revealed through indirect observations and it has been suggested that it reacts fast with surrounding water to form formaldehyde (Jensen et al., 2010).

H$_2$C•-O-SO$_3^-$ can also react fast with O$_2$ (Reaction R8) to form a peroxyl radical (•OO-CH$_2$-O-SO$_3^-$) through a barrierless process with the release of 54.83 kcal mol$^{-1}$ Gibbs free energy. Based on the above on the chemistry of peroxyl radicals, two molecules of •OO-CH$_2$-O-SO$_3^-$ can combine to form a tetroxide, which can further decompose to generate •O-CH$_2$-O-SO$_3^-$ by overcoming a Gibbs free energy barrier of 14.12 kcal mol$^{-1}$ (see Fig. 5). The alkoxy radical •O-CH$_2$-O-SO$_3^-$ can readily decompose to form HSO$_4^-$ and HCO• by overcoming a barrier of 7.92 kcal mol$^{-1}$, and as explained above and exemplified in Fig. 5, its formation from H$_2$C•-O-SO$_3^-$ reaction with O$_2$ is less thermodynamically favorable than with O$_3$, despite the overall rate of the former should be higher than that of the latter due to the high atmospheric concentration of O$_2$. It follows from the above mechanisms that in most atmospherically relevant conditions, the pathway for •O-CH$_2$-O-SO$_3^-$ formation from the H$_2$C•-O-SO$_3^-$ reaction with O$_3$ can readily complement that from Bennett and Summers that involves reaction with O$_2$ (Bennett and Summers, 1974).

In addition to the mechanisms speculated by experimental studies (Kwong et al., 2018), the combination of mechanisms both in the gas phase and the aqueous phase presented in this study provides additional pathways for inorganic sulfate formation from the reaction of methyl sulfate with HO• radicals, namely the decomposition of H$_2$C•-O-SO$_3^-$ or H$_2$C•-O-

**Table 2.** Gibbs free energy changes ($\Delta G$) for all intermediate species in the reaction of deprotonated methyl sulfate with HO$^\bullet$ radicals at 298.15 K and 1 M. Energy units are kcal mol$^{-1}$. "RC" stands for intermediate reactant complex, "TS" stands for transition state, and "PD" stands for product.

| Species | $\Delta G$ |
| --- | --- |
| CH$_3$-O-SO$_3^-$ + $\bullet$OH $\leftrightarrow$ RC4 $\rightarrow$ TS4 $\rightarrow$ PD4 (H$_2$C$\bullet$-O-SO$_3^-$ $\bullet\bullet\bullet$H$_2$O) | |
| H$_2$C$\bullet$-O-SO$_3^-$ $\bullet\bullet\bullet$H$_2$O $\rightarrow$ TS4-b $\rightarrow$ PD4-b (H$_2$CO$\bullet\bullet\bullet$SO$_3\bullet^-$ $\bullet\bullet\bullet$H$_2$O) | |
| RC4 | 6.07 |
| TS4 | 9.61 |
| PD4 | −14.28 |
| TS4-b | 4.62 |
| PD4-b | −17.34 |
| H$_2$C$\bullet$-O-SO$_3^-$ + O$_3$ $\leftrightarrow$ RC5 $\rightarrow$ TS5 $\rightarrow$ PD5 ($\bullet$O-H$_2$C-O-SO$_3^-$ + O$_2$) | |
| $\bullet$O-H$_2$C-O-SO$_3^-$ (RC5-b) $\rightarrow$ TS5-b $\rightarrow$ PD5-b (HSO$_4^-$ $\bullet\bullet\bullet$HCO$\bullet$) | |
| RC5 | −52.05 |
| TS5 | −58.03 |
| PD5 | −59.64 |
| RC5-b | −44.72 |
| TS5-b | −36.80 |
| PD5-b | −60.73 |
| H$_2$C$\bullet$-O-SO$_3^-$ + O$_2$ $\rightarrow$ $\bullet$OO-CH$_2$-O-SO$_3^-$ (PD6) | −54.83 |
| $\bullet$OO-CH$_2$-O-SO$_3^-$ + $\bullet$OO-CH$_2$-O-SO$_3^-$ $\leftrightarrow$ RC7 $\rightarrow$ TS7 $\rightarrow$ PD7 $\rightarrow$ $\bullet$O-CH$_2$-O-SO$_3^-$ | |
| RC7 | −103.58 |
| TS7 | −89.46 |
| PD7 | −141.39 |
| HOOC-CH$_2$-O-SO$_3^-$ + $\bullet$OH $\leftrightarrow$ RC8 $\rightarrow$ TS8 $\rightarrow$ PD8 (HOOC-CH$\bullet$-O-SO$_3^-$ $\bullet\bullet\bullet$H$_2$O) | |
| HOOC-CH$\bullet$-O-SO$_3^-$ (RC9) $\rightarrow$ TS9 $\rightarrow$ PD9 (HOOC-CHO$\bullet\bullet\bullet$SO$_3\bullet^-$) | |
| RC8 | 5.61 |
| TS8 | 11.52 |
| PD8 | −18.08 |
| RC9 | −5.69 |
| TS9 | 12.98 |
| PD9 | −3.66 |
| HOOC-CH$\bullet$-O-SO$_3^-$ + O$_3$ $\rightarrow$ O$_2$ + HOOC-CH(O)$\bullet$-O-SO$_3^-$ | −47.79 |
| HOOC-CH$\bullet$-O-SO$_3^-$ + O$_2$ $\rightarrow$ HOOC-CH(OO)$\bullet$-O-SO$_3^-$ | −43.01 |

SO$_3$H and their reactions with O$_3$. These processes can drive significant changes in the chemical composition of aerosol, especially in terms of sulfate mass loadings.

## 3.2 Reaction mechanism of glycolic acid sulfate with HO$\bullet$ radicals

Hydrogen abstraction from glycolic acid sulfate could occur both from -CH$_2$- and -COOH groups according to the fol-

lowing reactions:

$$\text{HOOC}-\text{CH}_2-\text{O}-\text{SO}_3\text{H} + \text{HO}\bullet \rightarrow \text{H}_2\text{O} + \text{HOOC}$$
$$-\text{CH}\bullet-\text{O}-\text{SO}_3\text{H}, \qquad (\text{R9})$$

$$\text{HOOC}-\text{CH}_2-\text{O}-\text{SO}_3\text{H} + \text{HO}\bullet \rightarrow \text{H}_2\text{O} + \bullet\text{OC(O)}$$
$$-\text{CH}_2-\text{O}-\text{SO}_3\text{H}. \qquad (\text{R10})$$

The mechanism of Reaction (R9) is similar to that of the hydrogen abstraction from methyl sulfate that forms an alkyl radical. Through this process, glycolic acid sulfate readily undergoes a hydrogen abstraction from the -CH$_2$- group by HO$\bullet$, resulting in HOOC-CH$\bullet$-O-SO$_3$H formation. The reactant complex in this process lies at 4.65 kcal mol$^{-1}$ at 298.15 K and 1 atm, and the transition state for its conver-

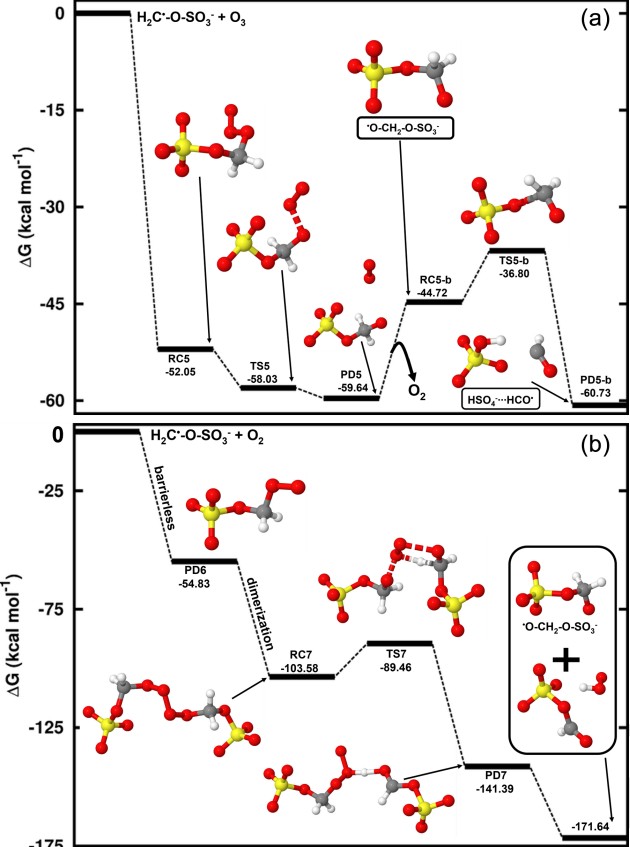

**Figure 5.** Gibbs free energy changes (in kcal mol$^{-1}$) and optimized structures for all intermediates in the reaction of $CH_2$•-O-SO$_3^-$ with O$_3$ **(a)** and O$_2$ **(b)**. The sulfur atom is yellow, the oxygen atom is red, the carbon atom is gray, and the hydrogen atom is white.

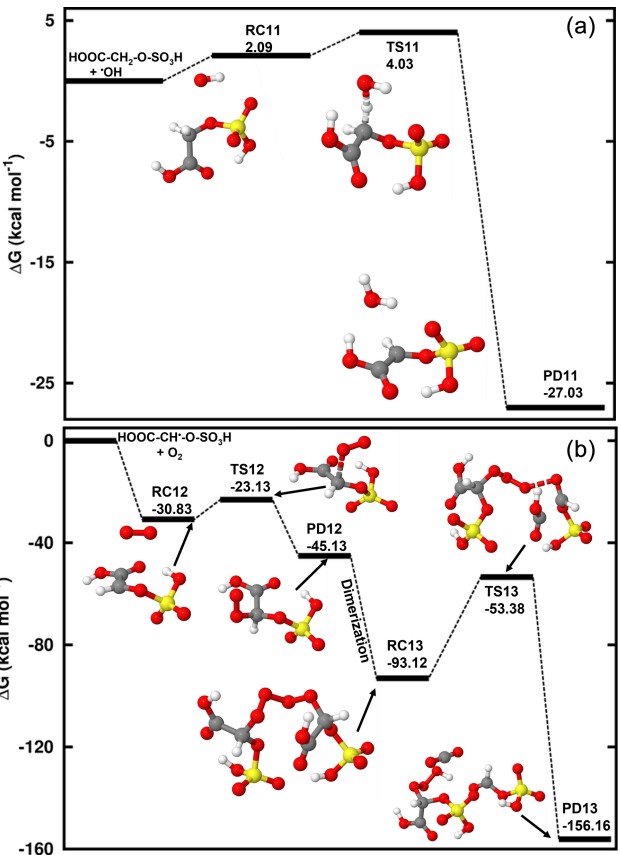

**Figure 6.** Gibbs free energy changes (in kcal mol$^{-1}$) and optimized structures for all intermediates in the reaction of HOOC-CH$_2$-O-SO$_3$H reaction with HO• radicals proceeding through hydrogen abstraction from the -CH$_2$- group **(a)** and in the HOOC-CH•-O-SO$_3$H reaction with O$_2$ **(b)**. The sulfur atom is yellow, the oxygen atom is red, the carbon atom is gray, and the hydrogen atom is white.

sion is located 2.13 kcal mol$^{-1}$ above the reactant complex. We determined a bimolecular rate constant of the overall reaction to be $6.17 \times 10^{-12}$ cm$^3$ molec.$^{-1}$ s$^{-1}$ for this re-action at 298.15 K. This shows that hydrogen abstraction by •OH from glycolic acid sulfate is more favorable than from methyl sulfate, hereby highlighting the enhancing ef-fect of the carboxyl substituent. The further chemistry of HOOC-CH•-O-SO$_3$H is examined through reactions with O$_3$ and O$_2$. Our calculations show that contrary to the re-action with H$_2$C•-O-SO$_3$H, O$_3$ hardly reacts with HOOC-CH•-O-SO$_3$H as the O$_3$•••HOOC-CH•-O-SO$_3$H formation is highly endergonic at standard conditions. However, the reac-tion with O$_2$ is seen to be fast, proceeding through formation of the reactant complex that is readily converted to HOOC-CH(OO)•-O-SO$_3$H. The observed negative Gibbs free en-ergy barrier ($-9.02$ kcal mol$^{-1}$ below the reactant complex) in this conversion indicates that the formation of HOOC-CH(OO)•-O-SO$_3$H is almost instantaneous at standard con-ditions. Two molecules of HOOC-CH(OO)•-O-SO$_3$H de-velop into a tetroxide that then decomposes to HC(O)-O-

SO$_3$H and HOOC-CH(O)•-O-SO$_3$H. The energetics of this reaction are provided in Fig. 6 and Table S5.

The abstraction from -COOH led to •OC(O)-CH$_2$-O-SO$_3$H (Reaction R10) that further decomposes to methy-lene sulfate radical (H$_2$C•-O-SO$_3$H) and CO$_2$. The struc-tures and energetics of all intermediate states of this re-action are given in Fig. S5 and Table S6. At the same level of theory, we determined a biomolecular rate constant of $3.86 \times 10^{-14}$ cm$^3$ molec.$^{-1}$ s$^{-1}$, two orders of magnitude lower than the hydrogen abstraction from the -CH$_2$- group. This indicates that hydrogen abstraction from glycolic acid would follow two competitive pathways although the path-way leading the alkyl radical would be somewhat preferred. It can be inferred that for organosulfates that have a -COOH substituent at the $\beta$-position relative to the sulfate group, de-carboxylation would be a possible outcome of their decom-position. The chemistry of H$_2$C•-O-SO$_3$H was assessed in Sect. 3.1 above.

We further investigated the HO•-initiated reaction of gly-colic acid sulfate in the aqueous phase, where the deproto-

nated state, $HOOC-CH_2-O-SO_3^-$, is predominant. The preliminary interaction in this reaction is similar to that in the reaction of $CH_3-O-SO_3^-$, with HO• abstracting the hydrogen atom from the $-CH_2-$ group to form the product complex, $HOOC-CH•-O-SO_3^- \cdots H_2O$. The energies and structures of all different stationary states of this reaction are given in Fig. 7, while further energetics details are provided in Table 2. A bimolecular rate constant of $7.29 \times 10^8 \, M^{-1} \, s^{-1}$ is determined for this reaction, in good agreement with the experimental report by Buxton et al. (1988). $HOOC-CH•-O-SO_3^- \cdots H_2O$ formation is highly exergonic, with $-18.69 \, kcal \, mol^{-1}$ Gibbs free energy at 298.15 K and 1 M. Its decomposition back to the initial reactants is prevented by a substantially high energy barrier of $29.60 \, kcal \, mol^{-1}$. Based on this backward process, an atmospheric lifetime of $6.30 \times 10^8 \, s$ is predicted for $HOOC-CH•-O-SO_3^- \cdots H_2O$ under relevant atmospheric conditions, long enough for this product complex to be subject to collisions with nearly all relevant atmospheric oxidants. $H_2O$ further dissociates from the product complex, leaving bare $HOOC-CH•-O-SO_3^-$ to undergo the following decomposition processes:

$$HOOC-CH•-O-SO_3^- \rightarrow HOOC-CHO + SO_3^{•-}, \quad (R11)$$

$$HOOC-CH•-O-SO_3^- + O_3 \rightarrow O_2 \\ + HOOC-CH^•(O)-O-SO_3^-, \quad (R12)$$

$$HOOC-CH•-O-SO_3^- + O_2 \\ \rightarrow \rightarrow HOOC-CH^•(O)-O-SO_3^-. \quad (R13)$$

Following Reaction (R11), $HOOC-CH•-O-SO_3^-$ can undergo $O-SO_3$ bond cleavage and form the $HOOC-CHO \cdots SO_3^{•-}$ product complex at a unimolecular rate constant of $1.24 \times 10^{-1} \, s^{-1}$. The electronic charge analysis (shown in Fig. S6a) confirms the distribution of the unpaired electron on $SO_3$ while $HOOC-CHO$ is electrically neutral. Knowing that $SO_3^{•-}$ has no other atmospheric chemical fate than inorganic sulfate, it follows that glycolic acid sulfate transformation by a HO•-initiated reaction would produce glyoxylic acid and sulfate at a nearly equal rate constant as $CH_2-O-SO_3^-$ for a similar mechanism. The significance of this reaction will, however, depend on the rates of $HOOC-CH•-O-SO_3^-$ reactions via other pathways.

Similar to $H_2C•-O-SO_3^-$, the $HOOC-CH•-O-SO_3^-$ reaction with $O_3$ is completely downhill and directly undergoes an oxygen atom transfer for $O_3$ to the $-CH-$ group, forming the alkoxy radical, $HOOC-CH(O)•-O-SO_3^-$ (Reaction R12). $HOOC-CH(O)•-O-SO_3^-$ is susceptible to further decompose to $CO_2 + HCO• + HSO_4^-$. However, we were unable to locate the appropriate transition state, which is seemingly associated with the mesomeric effect induced by the presence of an unpaired electron and entertained by the $-COOH$ function. This situation was not observed in the decomposition of $•O-CH_2-O-SO_3^-$.

Another reaction pathway for $HOOC-CH•-O-SO_3^-$ reaction is by $O_2$ addition (Reaction R13) in a barrierless process to form a peroxyl radical ($HOOC-CH(OO•)-O-SO_3^-$) with the release of $43.01 \, kcal \, mol^{-1}$ Gibbs free energy. $HOOC-CH(OO•)-O-SO_3^-$ can recombine with each other to form a tetroxide ($O_3^- S-O-CH(COOH)-OOOO-CH(COOH)-O-SO_3^-$) as shown in Fig. 7. However, we found that contrary to the case of $•OO-CH_2-O-SO_3^-$, where the tetroxide could readily decompose to form the $•O-CH_2-O-SO_3^-$ radical in the singlet state, the fragmentation of $O_3^- S-O-CH(COOH)-OOOO-CH(COOH)-O-SO_3^-$ to form the alkoxyl radical $HOOC-CH(O)•-O-SO_3^-$ occurred on the triplet state instead. The triplet electronic state was shown from a recent study to be favorable to the decomposition of tetroxides to alkoxyl radicals from some atmospherically relevant peroxyl radicals (Salo et al., 2022). Figure S6b clearly shows the antibonding orbitals in the triplet state prior to the formation of $HOOC-CH(O)•-O-SO_3^-$ upon which rests the unpaired electron.

## 3.3 Atmospheric implications

Organosulfates are important organic tracers for aerosols in the atmosphere. Although sufficient information on their sources and abundance has been gathered from previous studies, the understanding of the mechanisms of their transformation in the atmosphere remains incomplete. By investigating the decomposition mechanisms of two small atmospheric organosulfates (methyl sulfate and glycolic acid sulfate) by reaction with HO• radicals in this study, it was shown that the reaction of glycolic acid sulfate in the gas phase is more kinetically favorable than that of methyl sulfate, which can be attributed to the effect of $-COOH$ substitution that stabilizes the intermediate reactant complex from glycolic acid. The chemical transformation of both organosulfates was seen to be more effective in the aqueous phase, with the reaction of methyl sulfate being more extensive than that of glycolic acid sulfate. The main products of these transformations are carbonyl compounds and inorganic sulfate, for which detailed mechanisms are provided. Not only does this study clarify the effect of substituents on the fragmentation of organosulfates, but it also complements previous experimental observations on methyl sulfate oxidation by HO• radicals (Kwong et al., 2018). For this reaction, we obtained a rate constant of $1.14 \times 10^{-13} \, cm^3 \, molec.^{-1} \, s^{-1}$ in the gas phase, in agreement with the previously reported experimental value ($(3.79 \pm 0.19) \times 10^{-13} \, cm^3 \, molec.^{-1} \, s^{-1}$) (Kwong et al., 2018). We also report a rate bimolecular rate constant of $6.17 \times 10^{-12} \, cm^3 \, molec.^{-1} \, s^{-1}$ for the gas-phase reaction of HO• reaction with glycolic acid at 298.15 K. In the aqueous phase, we determined rate constants of $7.87 \times 10^6$ and $7.29 \times 10^8 \, M^{-1} \, s^{-1}$ for the reaction of methyl sulfate and glycolic acid sulfate, respectively.

Among the three processes investigated (self-decomposition, reaction with $O_3$, and reaction with $O_2$), alkoxyl radicals can be formed both from alkyl radicals reactions with $O_2$ and with $O_3$. From the discussion above on reaction mechanisms and energetics, we clarify that

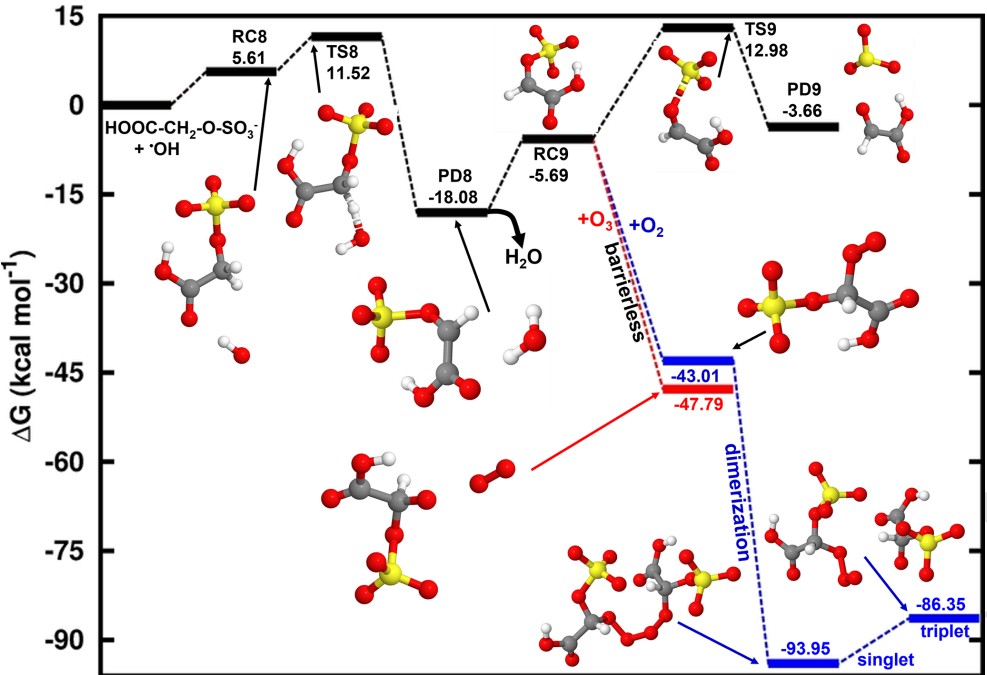

**Figure 7.** Gibbs free energy changes (in kcal mol$^{-1}$) and optimized structures for all stationary states in the reaction of HOOC-CH$_2$-O-SO$_3^-$ with HO• radicals, and subsequent reaction of the intermediate reactant complex with O$_3$ (red line) and O$_2$ (blue line). The sulfur atom is yellow, the oxygen atom is red, the carbon atom is gray, and the hydrogen atom is white.

besides the O$_2$ reaction, the reaction with O$_3$ could be a complementary intermediate step in the formation of alkoxy radicals that further decompose to inorganic sulfate and carbonyl compounds. However, considering real atmospheric conditions (with O$_2$ concentrations much higher than O$_3$ concentrations), this formation will be kinetically driven by the reaction with O$_2$, whereas the reaction with O$_3$ may exclusively become relevant in environments highly enriched with ozone.

Overall, the kinetics results show a moderate difference between the rate constants of HO• reactions with methyl sulfate and glycolic acid sulfate. The slightly high rate constant of the reaction of glycolic acid sulfate indicates the enhancing effect of the -COOH group in the hydrogen abstraction by HO•. Moreover, we found that the hydrogen abstraction from the -COOH group in glycolic acid sulfate leads to decarboxylation and eventually forms similar products as methyl sulfate. It can be inferred that for organosulfates having a carboxyl substituent at the $\beta$-position relative to the sulfate group, decarboxylation would take place, leading to the formation of the corresponding alkyl sulfate radical. This highlights the potential role that chemical substitution on the carbon chain of organosulfates may play during their decomposition. Given the high variety of organosulfates detected in atmospheric particles, it is necessary to deeply evaluate the role of molecular structures in their chemical transformation in order to guarantee proper understanding of their impact on the chemical composition of aerosols. A general trend

of the effect of chemical substitution can be obtained from segregated studies of the chemical transformation of different classes of organosulfates derived from anthropogenic and biogenic precursors.

**Data availability.** All data from this research can be obtained upon request by contacting the corresponding author.

**Supplement.** The supplement related to this article is available online at [the link will be implemented upon publication].

**Author contributions.** Conceptualization: NTT; funding acquisition: NTT and LD; investigation: LX and NTT; supervision: LD; writing – original draft preparation: LX and NTT. writing – review and editing: SNT, JNG, and LD.

**Competing interests.** The contact author has declared that none of the authors has any competing interests.

ery effort to include appropriate place names, the final responsibility lies with the authors.

**Acknowledgements.** The authors acknowledge the Wuxi Hengding Supercomputing Center Co., LTD and National Supercomputer Center in Tianjin for providing the computational resources. We thank Kristian H. Møller and Rasmus V. Otkjær of the University of Copenhagen for providing the script to calculate the Eckart tunneling correction.

**Financial support.** This research has been supported by the National Key Research and Development Program of China (grant no. 2023YFC3706203), the National Natural Science Foundation of China (grant no. 22376121), and the Natural Science Foundation of Shandong Province (grant no. ZR2023MD041).

**Review statement.** This paper was edited by Theodora Nah and reviewed by three anonymous referees.

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
