# Peer review of "Atmospheric fate of organosulfates through gas-phase and aqueous-phase reaction with hydroxyl radicals: implications in inorganic sulfate formation"

_EGUsphere, 2025_

## Author Response (AR3)

**Author response to egusphere-2025-29**

We thank the Referees and the Editor for their insightful comments on our manuscript. Below, we provide point-to-point response to all the comments. For clarity, the Referees' comments are reproduced in blue color text, authors' replies are in black color and modifications to the manuscript are in red color text.

**Reply to Anonymous Referee #1**

In this work the authors carried out quantum calculations to explore the gas-phase and aqueous-phase reactions of organosulfate (namely methyl sulfate and glycolic acid sulfate) with dissolved OH radicals. The simulated results clearly demonstrated the main reaction outcome is formation of inorganic sulfate and carbonyl compounds. At the same time, the results show that the nature of the substituent have some impact on the reaction mechanism. The potential reaction mechanisms were discussed in details and in general agreed with the existing experimental results. The paper is well written and provides greater mechanistic understanding of the chemistry of organosulfates in the aspect of inorganic sulfate formation. This work fits well to the scope of this journal and I recommend publication in Atmospheric Chemistry and Physics, after the following comments have been addressed.

1 What are the atmospheric concentrations of the reactants considered in this study? Do the authors think the reactants concentrations would affect the reaction mechanism?

By its principle, quantum chemical (QC) calculations allow to solve for the electronic configuration of the reacting system and provide the reaction energies at all reaction states including the transition states, regardless of the reactants concentrations, and to analyze the reaction mechanisms of the developed reactions. This approach specially focuses on the electronic interaction between the reactants in order to provide atomic level insights into the processes driving the reaction. In this regard, QC calculations are mainly designed to predict and explain unknown reactions and are not affected by the reactants concentrations. Likewise, the transition states obtained from QC calculations are not dependent on the reactants concentrations and hence, the latter do not affect the reaction rate constants.

By default, the rate constant of a reaction in the gas-phase (and aqueous phase, respectively) (given by Eq. (3) in the main manuscript) is determined from QC calculations at a standard pressure of 1 atm (and standard concentration of 1 M, respectively), with all other parameters being independent of the reactants concentrations. While the reactants concentrations do not specifically affect the rate constants, they alter the overall kinetics by affecting the reaction rates, which depend on the reactants concentrations. The rate constant is a fundamental parameter in kinetics that represents the constant of proportionality relating the reaction rate to the concentrations of reactants, while the reaction rate measures the change in reactant or product concentration per unit time.

The mention of standard concentration of 1 atm in the gas-phase and 1 M in the aqueous-phase for computing the thermochemistry and the rate constant has already been made in Section 2.2 of the main manuscript.

In addition to the above explanation, the following is added in the revised manuscript to highlight the atmospheric concentrations of methyl sulfate and glycolic acid sulfate.

Line 65

"Both organosulfates have been detected at various locations around the world at concentrations in the ranges $1.08 \times 10^6 - 5.01 \times 10^7$ molecule cm-3 for methyl sulfate (Hettiyadura et al., 2015; Peng et al., 2021) and $1.16 \times 10^7 - 4.71 \times 10^8$ molecule cm-3 for glycolic acid sulfate (Huang et al., 2018; Hettiyadura et al., 2015; Wang et al., 2021; Hughes and Stone, 2019; Cai et al., 2020; Liao et al., 2015)."

2 The authors highlight that chemical substitution has some impact on the course of the reaction. Is there a general rule regarding this?

Different functional groups can exhibit different reactivities depending on the nature of the substituents and actual environmental conditions. However, as already stated in Section 3.3 of the manuscript, understanding the segregated impact of different substituents on the chemical transformation of organosulfates necessitates a deeper evaluation of the role of molecular structures in their transformation. Given the large variety of organosulfates in the atmosphere, more studies on their chemical transformation need to be conducted before a general trend on the effects of chemical substitution can be observed.

Related to this, the following text has been added in the revised manuscript at line 367:

"A general trend of the effect of chemical substitution can be obtained from segregated studies of the chemical transformation of different classes of organosulfates derived from anthropogenic and biogenic precursors."

3 How do the authors justify their choice of the M062X/6-311+g(2df,2pd) method for geometry optimizations and frequency calculations in the studied reactions?

While the M062X density functional has been shown in previous studies to reliably describe atmospherically relevant reactions and efficiently resolve the transitions states (*J. Chem. Theory Comput.* 8, 6, 2071-2077, **2012**; *Phys. Chem. Chem. Phys.* 15, 16442-16445, **2013**; *J. Phys. Chem. A* 117, 30, 6695-6701, **2013**), the 6-311+g(2df,2pd) basis set is complete enough to satisfactorily account for polarization functions and diffuse functions to describe weak interactions. The M062X/6-311+g(2df,2pd) combination has successfully been used in a series of previous studies. Although this has been clarified in the manuscript, the sentence at line 83 has been modified in the revised manuscript as follows to further justify our choice

"The M062X/6-311+g(2df,2pd) method has successfully been used in a series of previous studies (Ding et al., 2023; Wang et al., 2024; Cheng et al., 2022) and proved

to efficiently resolve the transition state configuration. Given the size of the system investigated here, this level of theory appears to be a good compromise between accuracy and computation time."

4 In Figure 6, the mechanism from the intermediate lying at -43.01 kcal/mol to the product at -93.95 kcal/mol is not clear. Can the authors elucidate further on this?

The reaction step lying from -43.01 kcal mol-1 to -93.95 kcal mol-1 in Figure 6 corresponds to the dimerization of the peroxy radical (HOOC-CH(OO•)-O-SO$_3^-$) occurring through a barrierless process. This process was described in the manuscript, and it is now labelled as "dimerization" in Figure 6 in the revised manuscript.

[Figure]

**Figure 7**: Gibbs free energy changes (in kcal mol$^{-1}$) and optimized structures for all stationary states in the reaction of HOOC-CH$_2$-O-SO$_3^-$ with HO• radicals, and subsequent reaction of the intermediate reactant complex with O$_3$ (red line) and O$_2$ (blue line). The sulfur atom is in yellow, the oxygen atom is in red, the carbon atom is in grey and the hydrogen atom is in white color.

5 In regards of the high atmospheric concentration of O$_2$, it is good that they authors consider this as an important reactant in the intermediate steps of the reactions in the aqueous-phase. However, why didn't they examine the reaction of this oxidant (O$_2$) in the intermediate steps of the reaction in the gas-phase?

Despite previous attempts to investigate O$_2$ reaction with the immediate radical (H$_2$C•-O-SO$_3$H···H$_2$O) resulting from gas-phase OH reaction with the organosulfate failed initially, this system has been reconsidered without additional water molecule. Our calculations indicate that H$_2$C•-O-SO$_3$H would react fast with O$_2$ to form a peroxy radical (•OO-CH$_2$-O-SO$_3$H). Thereafter, two molecules of this radical would combine to form a dimer, which can further decompose to generate formic sulfuric anhydride

(HOC-O-SO$_3$H). The following changes have been made in the revised manuscript to highlight the outcome of the O$_2$ + H$_2$C•-O-SO$_3$H reaction.

The sentence at line 19 is modified to the following:

"The primary reaction products are inorganic sulfate, carbonyl compounds and formic sulfuric anhydride."

The sentence at Line 21 is modified to the following:

"Additionally, while prior studies suggested O$_2$ as primary oxidant in the fragmentation of organosulfates, this study unveils O$_3$ as a complementary oxidant in the intermediate steps of this process."

The following reaction is added at Line 155:

$$H_2C•\text{-O-}SO_3H + O_2 \rightarrow \rightarrow HC(O)\text{-O-}SO_3H , \hspace{3cm} (R5)$$

The following text is added at Line 222

Contrary to the self-decomposition of H2C•-O-SO3H and reactions with NO2 and O3 that are favorable with H2C•-O-SO3H hydrates, we were unable to fully optimize the O2 reaction with H2C•-O-SO3H hydrates but with the unhydrated system, instead. This led to exergonic formation of the •OO-CH2-O-SO3H peroxyl radical with 45.98 kcal mol-1 Gibbs free energy gain. The chemistry of peroxyl radicals has been the subject of several experimental and theoretical studies. It is widely accepted that the peroxyl radical would predominantly decompose to form alkoxyl radicals or alcohols along with carbonyl compounds through tetroxide intermediates (Russell, 1957). However, although the end products from this decomposition have been verified experimentally, the mechanisms have often been deemed unlikely due to inconsistency between thermodynamic experiments and computational studies (Nangia and Benson, 1980; Zhang et al., 2012; Liang et al., 2011). Moreover, the effort to elucidate alcohols and carbonyl compounds formation from the decomposition of the tetroxide in a previous study could not be achieved due to the impossibility to determine the corresponding transition states (Ghigo et al., 2003), while alkoxyl radical formation was observed to simply correspond to the dissociation of the tetroxide. A most recent theoretical study specifically focusing on the decomposition pathways of the tetroxide intermediate indicated that although substantial uncertainties may exist in their computed energetics, alkoxyl radicals are likely primary products from atmospherically relevant peroxyl radicals (Salo et al., 2022). Following the above reasoning, our calculations indicate that two molecules of •OO-CH2-O-SO3H recombine to form a tetroxide that can further decompose to generate two alkoxyl radicals •O-CH2-O-SO3H clustered to molecular oxygen. Then, the two •O-CH2-O-SO3H radicals quickly interact with O2 to form H2O2 and formic sulfuric anhydride (HC(O)-O-SO3H) (reaction (R5)). The latter product has been identified to enhance new particle formation (An, 2024). Energetics and structures of all intermediates in this reaction are given in Figure 3.

6 Line 130. Can the authors use a different word than "achieved" at the end of the sentence?

The sentence was re-written as follows:

"This is likely because for this configuration, the minimum amount of water (2 molecules) necessary to achieve the stability of $H_3O^+$ in the gas-phase has been reached (Markovitch and Agmon, 2007; Heindel et al., 2018)."

7 In the caption of Figure 2, it is likely that product (C) is clustered to 3 water molecules.

The caption for Figure 2 has been corrected to:

**Figure 2: Representation of the spin density (blue color) on the electronic states of the decomposition of (A) $H_2C•-O-SO_3H\cdots(H_2O)_3$ and (B) $•O-H_2C-O-SO_3H$. From left to right are the reactant, transition state and the product complex, respectively. The sulfur atom is in yellow, the oxygen atom is in red, the carbon atom is in grey and the hydrogen atom is in white.**

**Reply to Anonymous Referee #2**

Manuscript review " Atmospheric fate of organosulfates through gas-phase and aqueous-phase reaction with hydroxyl radicals: implications in inorganic sulfate formation" for EGUSPHERE.

In this manuscript, the authors report on the oxidation of organosulfates, specifically methyl sulfate and glycolic acid sulfate, by OH radicals in the gas phase and in the aqueous phase. The investigations were carried out using DFT calculations with GAUSSIAN, whereby in addition to the electronic structure and thermochemistry, the reaction kinetics of the oxidation reaction and the respective intermediates were also calculated.

The questions and comments on the manuscript are listed below.

In the abstract, the authors should make it clear that they have carried out the DFT calculations.

The second sentence in the Abstract was revised as follows to highlight the use of DFT calculations in this study.
This study uses quantum chemical calculations based on density functional theory to explore the reactions of some prevalent organosulfates, specifically methyl sulfate and glycolic acid sulfate, with hydroxyl radicals (HO•) in the gas-phase and aqueous-phase.

Please check your manuscript for colloquial language, e.g. in line 17 "unfriendly". A molecule does not have these attributes.

In our reply to another comment of the Referee, we conducted more calculations for glycolic acid and based on the results, the sentences at lines 17-19 were deleted.

Line 16: How likely is the reaction of the alkyl radical with ozone (O3) in the presence of oxygen (O2), if we assume that an ozone concentration in the range of 10 to 100 ppb compared to 20% oxygen in the atmosphere?

It is obvious that the reaction of alkyl radicals with $O_3$ in the presence of $O_2$ is generally not highly favorable compared to both the reaction with $O_2$ and the $O_3$ reaction with other possible radicals. Concisely, under normal atmospheric conditions, alkyl radicals will overwhelmingly react with $O_2$ to form peroxy radicals rather than reacting with $O_3$. The later reaction may have some impact, though still limited, under elevated $O_3$ conditions such as in urban areas, exclusively. Owing to this argument, the related sentence in the revised manuscript has been modified to highlight the preference for alkyl radicals to react primarily with $O_2$ rather than $O_3$.

Line 15

Results indicate that all reactions initiate with hydrogen abstraction by HO• from CH3- in methyl sulfate and from -CH2- and -COOH in glycolic acid sulfate, followed by the further reaction of the resulting radicals through self-decomposition interaction with $O_2$ and, possibly, $O_3$.

Line 20: Why only the rate constant of OH radicals with methyl sulfonate is shown?

For comparison purposes, we initially reported only the rate constant value of the OH reaction with methyl sulfate for which the experimental data is available. In the revised manuscript, we also report the rate constant for the OH reaction with glycolic acid sulfate.

The sentence at line 20 is revised to:

Rate constants of $1.14 \times 10^{-13}$ cm$^3$ molecule$^{-1}$ s$^{-1}$ and $6.17 \times 10^{-12}$ cm$^3$ molecule$^{-1}$ s$^{-1}$ at 298.15 K were determined for the gas-phase reaction of methyl sulfate and glycolic acid sulfate, respectively. The former value is consistent with a previous experimental report.

The following sentence is added at line 351

We also report a rate bimolecular rate constant of $6.17 \times 10^{-12}$ cm$^3$ molecule$^{-1}$ s$^{-1}$ for the gas-phase reaction of HO• reaction with glycolic acid at 298.15 K.

Why does the reader need the text from line 101 to line 113? Because none of the diffusion parameters are discussed later in the manuscript text.

This part of the manuscript provides details for calculating the bimolecular rate constants for aqueous-phase reactions. Numerical details for diffusion parameters are now provided in the revised Supplement and the following text is added in the revised manuscript at line 113:

All numerical values for radii, diffusion coefficients of reactants and the steady-state Smoluchowski rate constants are provided in Tables S1 and S2 in the Supplement.

Line 134: How would the unimolecular rate constant develop if the number of water molecules corresponds to the number of water molecules in the hydration shell in the aqueous phase?

From our calculations, the studied organosulfates would rarely bind with more than two water molecules in the gas-phase at ambient conditions. For methyl sulfate for example, we have reported unimolecular rate constants of $1.15 \times 10^8$ s$^{-1}$, $7.89 \times 10^7$ s$^{-1}$ and $1.59 \times 10^{10}$ s$^{-1}$, for the unhydrated, monohydrated and dehydrated reactions, respectively. These values exhibit a non-uniform trend, with a moderate decrease from the unhydrated reaction to the monohydrated reaction, followed by an increase from the monohydrated reaction to the dehydrated reaction. Although the possibility of the number of binding water molecules in the gas-phase reaching the number of water molecules in the hydration shell in the aqueous-phase is highly improbable under relevant atmospheric temperature, pressure and humidity, if such scenario would occur, then the organosulfate would be deprotonated and its reaction would be comparable to the reaction in the aqueous-phase. In this regard, the aqueous-phase reaction was fully explored in **Section 3.1.2.**

Line 134: "Unimolecular rates constants" should read "Unimolecular rate constants"

This has been corrected.

Line 144: Could the authors show the rate constants instead of the lifetimes?

The lifetimes have been replaced by rate constants, and the related text at lines 144-148 has been further revised as follows:

Line 144

The stability of $H_2C\bullet$-O-SO$_3$H$\cdots$(H$_2$O)$_{n+1}$, examined relative to decomposition through the reverse reaction back to the HO$\bullet\cdots$CH$_3$-O-SO$_3$H$\cdots$(H$_2$O)$_n$ complex, reveals barriers heights of 28.64, 24.92 and 24.55 kcal mol$^{-1}$ for n = 0, 1, and 2, corresponding to unimolecular rate constants of $6.16\times10^{-9}$ s$^{-1}$, $3.30\times10^{-6}$ s$^{-1}$ and $6.14\times10^{-6}$ s$^{-1}$, respectively. These low-rate constants indicate that once formed, $H_2C\bullet$-O-SO$_3$H$\cdots$(H$_2$O)$_{n+1}$ would not react back to form initial reactants before possible collision with atmospheric oxidants have occurred.

Beside the uni-molecular decay rate, what would be the rate constant for the alkyl radical - alkyl radical recombination reaction in the absence of any other reaction partner? That would be interesting to know?

We assessed the radical-radical interaction both for $H_2C\bullet$-O-SO$_3$H and HOOC-CH$\bullet$-O-SO$_3$H, and found that the recombination occurs through C-C bond formation via a barrierless process. Respective resulting products are shown in **Figure R1** below. This recombination can be assumed to occur at collision rate, and would be the most likely fate of $H_2C\bullet$-O-SO$_3$H and HOOC-CH$\bullet$-O-SO$_3$H in the absence of other reaction partners such as O$_2$, O$_3$, or NO$_2$. Although the direct relevance of the resulting products is not clearly identified, their fate would mostly depend on reactions initiated by hydrogen abstraction from -CH$_2$- group for methylene sulfate and -CH- and/or -COOH groups for the radical resulting from glycolic acid sulfate.

The following text is added in the revised manuscript to highlight the outcome of radical-radical recombination at Line 222:

Considering the recombination of two $H_2C\bullet$-O-SO$_3$H molecules as another likely fate of $H_2C\bullet$-O-SO$_3$H the absence of other reaction partners such as O$_2$, O$_3$, or NO$_2$ besides self-decay, our preliminary calculations indicate that this recombination would occur fast and would proceed through C-C bond formation via a barrierless process. However, the relevance of the resulting product is not clearly established and its fate may mostly depend on reactions initiated by a hydrogen abstraction from the -CH$_2$- group.

[Figure]

**Figure R1: Alkyl radical recombination products from methyl sulfate (left) and glycolic acid sulfate (right). The sulfur atom is in yellow, the oxygen atom in red, the carbon atom is in grey and the hydrogen atom is in white.**

Line 152: How meaningful is the investigation of the reaction of the alkyl radical with ozone and NO2 in term of their solubility in water? Considering the self-decay reaction as well as the reaction with O2.

$NO_2$ dissolves effectively in water and it is known that gaseous $NO_2$ reacts with water to produce nitric acid, which is highly soluble in water (Ford and Miranda, 2020; Tan and Piri, 2013; Lee and Schwartz, 1981; England and Corcoran, 1974). In this regard, to explore the alkyl radical reaction with $NO_2$ in aqueous-phase would be equivalent to explicitly assessing the reaction with $NO_3^-$. This reaction was not investigated in the aqueous-phase in the present study. Considering $O_3$, it is highly soluble in water and this solubility plays an important role in various processes although its stability can be readily affected by various factors including ozone concentration, pH and ultraviolet light (Lovato et al., 2009). Considering its behavior as a prospective oxidant, ozone can react via a direct reaction pathway involving molecular ozone or by an indirect route involving various highly reactive intermediates that arise from its decomposition (Buehler et al., 1984; Buhler et al., 1984; Staehelin et al., 1984; Staehelin and Hoigne, 1982). Although we did not focus on reactions from indirect routes involving reactive intermediates that arise from $O_3$ decomposition, based on the above, the reaction of the alkyl radical with molecular $O_3$ is a likely process that would be of relative significance besides self-decay reaction and the reaction with $O_2$.

The sentence at lines 238-239 is modified to the following in the revised manuscript to justify the relevance of the reaction of alkyl radicals with $O_3$ and $O_3$.

While $NO_2$ dissolves effectively in water and is known to react with molecular water to produce nitric acid which is highly soluble in water (Ford and Miranda, 2020; Tan and Piri, 2013; Lee and Schwartz, 1981; England and Corcoran, 1974), the stability of $O_3$ solubility in water is readily affected by various factors including ozone concentration, pH and ultraviolet light (Lovato et al., 2009). Depending on environmental conditions, ozone can react via a direct reaction pathway involving molecular ozone or by an indirect route involving reactive intermediates that arise from its decomposition (Buehler et al., 1984; Buhler et al., 1984; Staehelin et al., 1984; Staehelin and Hoigne, 1982). Moreover, $O_3$ is known to be 13 times more soluble in water than $O_2$ (Seinfeld and Pandis, 1998). Hence, besides the self-decay reaction, the fate of $H_2C^\bullet\text{-O-SO}_3^-$ was examined via reactions with $O_2$ and $O_3$:

Figure 1: How reliable are the decimal places in the DFT calculations?

It is generally adopted in the quantum chemistry community that energy values from DFT calculations can be truncated to two decimal places.

Line 170: This statement depends on the ambient conditions, e.g. the concentration of organic substances in an aerosol particle. This is because the SO3.- could also react with organic substances. What are the intermediate steps in the formation of SO42- from SO3.-? Is there a difference in the gas phase or in the aqueous phase?

It is clear that the intermediate steps in sulfate formation from $SO_3^{\bullet-}$ in the gas-phase are different from those in the aqueous-phase. In the aqueous-phase, the mechanism of $SO_3^{\bullet-}$ conversion is relatively understood, with the major fate being its reaction with organic compounds to form sulfonates (Lv et al., 2021; Lv et al., 2024; Lai et al., 2024) while the gas-phase mechanism is clearly established. Several experimental studies showed that $SO_3^{\bullet-}$ would quickly hydrate in the gas-phase to form $SO_3^{\bullet-}\cdots(H_2O)_n$ cluster wherein the oxidation to sulfuric acid can occur according to the following reactions (Enghoff and Svensmark, 2008; Bork et al., 2013; Fehsenfeld and Ferguson, 1974; Svensmark et al., 2007; Tsona and Du, 2019).

$$SO_3\bullet^- + nH_2O \;\rightarrow\; SO_3\bullet^-\cdots(H_2O)_n\,, \qquad\qquad\qquad (R\text{-}R1)$$

$$SO_3\bullet^-\cdots(H_2O)_n \;\rightarrow\; H_2SO_4\cdots(H_2O)_{n-1} + e\,, \qquad\qquad (R\text{-}R2)$$

It was demonstrated from these studies that the free electron is involved in the stabilisation and formation of a cluster with at least one sulfuric acid molecule, wherein it acts as catalyst. These reactions are ion reactions with rate constants that are close to the time-scales of gas kinetic collision.

In the revised manuscript, the related text is modified to the following, to reinforce the understanding of the fate of $SO_3^{\bullet-}$ in the gas-phase

Line 170
Several experimental and theoretical studies showed that $SO_3^{\bullet-}$ would quickly hydrate in the gas-phase to form $SO_3^{\bullet-}\cdots(H_2O)_n$ cluster wherein the oxidation to sulfuric acid occurs in a mechanism stabilized and catalyzed by a free electron (Enghoff and Svensmark, 2008; Bork et al., 2013; Fehsenfeld and Ferguson, 1974; Svensmark et al., 2007; Tsona and Du, 2019). Considering this outcome for $SO_3^{\bullet-}$ in the gas-phase, $H_2CO$ and inorganic sulfate are expected products of the gas-phase reaction of methyl sulfate with HO• at ambient conditions.

Line 234: Do the authors have an explanation, why the obtained rate constant is approx. an order of magnitude lower compared to the measured value by Gweme and Styler, 2024 (doi: 10.1021/acs.jpca.4c02877) and the references therein?

While environmental factors such as pH, temperature and ionic strength can effectively alter the rate constants of aqueous-phase reactions from experimental studies, it is generally accepted that errors of 2-3 kcal mol$^{-1}$ in reaction energies may arise from DFT calculations (Bogojeski et al., 2020). These errors can induce up to $10^2$ increase or decrease in the reaction rate constants. Hence, it is likely that the difference between

rate constants obtained from our calculations and experimental values from Gweme and Styler (Gweme and Styler, 2024) arise from computational error besides possible experimental errors. The following text is added to the revised manuscript to justify the difference between the rate constant from our calculations and the experimental data.

Line 234
This value is about 13 times lower than a previous experimental value (Gweme and Styler, 2024), and the difference can be attributed both to computational errors and environmental factors.

Line 278: What would be the reaction barrier of the decomposition of the alkoxy radical to SO42- and formaldehyde?

The energy barrier of this decomposition is 7.92 kcal mol$^{-1}$. The sentence dealing with this decomposition is updated in the revised manuscript to include the reaction barrier.

Line 278
The alkoxy radical $\bullet O-CH_2-O-SO_3^-$ can readily decompose to form $HSO_4^-$ and $HCO\bullet$, by overcoming a barrier of 7.92 kcal mol$^{-1}$ and as explained above and exemplified in **Figure 5**, its formation from $H_2C\bullet-O-SO_3^-$ reaction with $O_2$ is less thermodynamically favorable than with $O_3$, despite the overall rate of the former can be higher than that of the latter due to the high concentration of $O_2$.

What is the authors' opinion regarding the decomposition of the peroxy radical similar to the decomposition of alkyl and alkoxyl radicals? Would that be possible?

Believed to be key intermediates in the oxidation of organic compounds, several mechanisms for the decomposition of peroxyl radicals are presented in the literature (Goldman et al., 2021; Salo et al., 2022; Tomaz et al., 2021). These radicals may undergo termination reactions or further autoxidation processes, which can yield a plethora of multifunctional products with low or extremely low saturation vapor pressures. In this study, we specifically investigated the pathway of two peroxy radicals recombining to for a tetroxide intermediate that further decomposes to form alkoxyl radicals.

To through more light on the outcome of peroxyl radicals, we added the following text in the revised manuscript.

Line 222
The chemistry of peroxyl radicals has been the subject of several experimental and theoretical studies. It is widely accepted that the peroxyl radical would predominantly decompose to form alkoxyl radicals or alcohols along with carbonyl compounds through tetroxide intermediates (Russell, 1957). However, although the end products from this decomposition have been verified experimentally, the mechanisms have often been deemed unlikely due to inconsistency between thermodynamic experiments and computational studies (Nangia and Benson, 1980; Zhang et al., 2012; Liang et al., 2011). Moreover, the effort to elucidate alcohols and carbonyl compounds formation from the decomposition of the tetroxide in a previous study could not be achieved due to the

impossibility to determine the corresponding transition states (Ghigo et al., 2003), while alkoxyl radical formation was observed to simply correspond to the dissociation of the tetroxide. A most recent theoretical study specifically focusing on the decomposition pathways of the tetroxide intermediate indicated that although substantial uncertainties may exist in their computed energetics, alkoxyl radicals are likely primary products from atmospherically relevant peroxyl radicals (Salo et al., 2022).

Line 291: Since the pKa of methyl sulfate is less than zero on the pH scale, and radicals generally have a lower pKa than the parent compound, how likely the presence of the distinct radical species is to be expect?

Lower pKa values are indicative of higher reactivities, which means that the radicals formed from hydrogen abstraction from organosulfates are expected to be more reactive that the organosulfates themselves. We have examined the possibilities of radicals conversions through many processes including self-decay, radical-radical combination as well as through reactions with $O_2$ and $O_3$, and found that radicals from both studied organosulfates react very fast with $O_2$ to form peroxyl radicals. Moreover, given the high abundance of atmospheric $O_2$, it is likely that reactions with $O_2$ will dominate the fate of radicals resulting from hydrogen abstraction from organosulfates. The chemistry of resulting peroxyl radicals has been carefully assessed in the revised manuscript.

Line 294: Why don't the authors give rate constants and lifetimes in the manuscript that are comparable to the discussion in the section on methyl sulfate? Why is there no discussion of the reactivity of similar compounds (glycolic acid) with OH to at least give an order of magnitude for the reactivity for glycolic acid sulfate?

We have now included some literature values of the rate constants both for the reactions of methyl sulfate and glycolic acid sulfate for comparison.

Line 312: How likely would be the decomposition of the alkyl radical to COOH and CHO-SO3-? What would be the energy barrier and the decomposition rate?

The configurational scans performed on $HOOC-CH\bullet-O-SO_3^-$ along the C-C coordinate showed that the decomposition along the C-C bond could not be achieved, probably because the targeted decomposition products (COOH and $HC-O-SO_3^-$) are not chemically stable. We found that the C-C bond could elongate from 1.45 Å to more than 3.40 Å, yet the energy of the system kept increasing. We conclude that this decomposition is not a likely fate for $HOOC-CH\bullet-O-SO_3^-$.

Line 317: Why the fate of SO3.- is discussed again?

The statement here was to highlight that one of the ultimate products of $HO\bullet$–initiated oxidation of glycolic acid sulfate would be inorganic sulfate through $SO_3^{\bullet-}$ intermediate.

Line 334: Could the authors provide information on the contribution of the three different decomposition pathways of tetroxide? It is known from the literature that about 10-20% of tetroxide in the aqueous phase is decomposed by the formation of molecular singlet oxygen and alkoxy radicals, depending on the parent compound. What would be the contribution of the H2O2 + RCRO (ketone) formation pathway and what would be the contribution of the RCROH (alcohol), RCRO (ketone) and O2

formation pathways? What would be the influence of the organic moiety of the molecules discussed here?

The chemistry of peroxyl radicals has been the subject of thorough investigation in many previous studies (Russell, 1957; Salo et al., 2022; Berndt et al., 2018; Tomaz et al., 2021; Goldman et al., 2021). Although no main channel exists for the reactions of peroxyl radicals, it has been established that their most relevant reactions can be categorized into propagating and terminating reactions. Earliest studies of this radical have led to the conclusion that the majority of peroxyl radical self- and cross-reactions either form alkoxyl radicals or alcohols along with carbonyl compounds through tetroxide intermediates according to the following processes (Russell, 1957; Lightfoot et al., 1992):

$$R\text{-}OO\bullet + R'\text{-}OO\bullet \rightarrow R\text{-}OOOO\text{-}R' \rightarrow R_{\bullet H}{=}O + R'\text{-}OH + {}^3O_2 , \qquad\qquad \text{(R-R3)}$$

$$\rightarrow R\text{-}OO\text{-}R' + {}^3O_2 , \qquad\qquad \text{(R-R4)}$$

$$\rightarrow R\text{-}O\bullet + R'\text{-}O\bullet + {}^3O_2 . \qquad\qquad \text{(R-R5)}$$

However, although the end products from the first study of this tetroxide decomposition have been verified experimentally, the mechanism itself has been deemed unlikely as it is inconsistent with thermodynamic experiments and computational studies (Nangia and Benson, 1980; Zhang et al., 2012; Liang et al., 2011).

The effort to elucidate the decomposition of the tetroxide into alcohol and carbonyl compound (**reaction (R-R3)**) was prevented by the impossibility to find the corresponding transition states (Ghigo et al., 2003), while alkoxyl radical formation (**reaction (R-R5)**) simply corresponds to the dissociation of the tetroxide. Furthermore, a most recent theoretical study specifically focusing on the decomposition pathways of the tetroxide intermediate indicated that although substantial uncertainties may exist in the computed energetics, alkoxyl radicals may primarily form from atmospherically relevant primary and secondary peroxyl radicals (Salo et al., 2022). Salo et al. also examined the reactions of peroxyl radicals with different organic moiety and found that although different organic moieties do not alter the nature of expected products, the barrier heights for the decomposition are affected both by the nature of the substituent and the carbon chain length (Salo et al., 2022).

We equally faced difficulties to locate the transition states in the formation of alcohol + carbonyl products, on the singlet surface as well as on the triplet surface, and our presented results are based on alkoxyl radical formation pathway, exclusively.

To clarify this, the sentence at line 222 is modified in the revised manuscript to the following:

The chemistry of peroxyl radicals has been the subject of several experimental and theoretical studies. It is widely accepted that the peroxyl radical would predominantly decompose to form alkoxyl radicals or alcohols along with carbonyl compounds through tetroxide intermediates (Russell, 1957). However, although the end products from this decomposition have been verified experimentally, the mechanisms have often been deemed unlikely due to inconsistency between thermodynamic experiments and

computational studies (Nangia and Benson, 1980; Zhang et al., 2012; Liang et al., 2011). Moreover, the effort to elucidate alcohols and carbonyl compounds formation from the decomposition of the tetroxide in a previous study could not be achieved due to the impossibility to determine the corresponding transition states (Ghigo et al., 2003), while alkoxyl radical formation was observed to simply correspond to the dissociation of the tetroxide. A most recent theoretical study specifically focusing on the decomposition pathways of the tetroxide intermediate indicated that although substantial uncertainties may exist in their computed energetics, alkoxyl radicals are likely primary products from atmospherically relevant peroxyl radicals (Salo et al., 2022).

Line 277

Based on the above on the chemistry of peroxyl radicals, two molecules of $\cdot OO\text{-}CH_2\text{-}O\text{-}SO_3^-$ can combine to form a tetroxide, which can further decompose to generate $\cdot O\text{-}CH_2\text{-}O\text{-}SO_3^-$ by overcoming a Gibbs free energy barrier of 14.12 kcal mol$^{-1}$ (see **Figure 5**).

The sentence at line 332 is modified to the following in the revised manuscript.

$HOOC\text{-}CH(OO\cdot)\text{-}O\text{-}SO_3^-$ can recombine with each other to form a tetroxide ($O_3^-S\text{-}O\text{-}CH(COOH)\text{-}OOOO\text{-}CH(COOH)\text{-}O\text{-}SO_3^-$) as shown in **Figure 6**. However, we found that contrary to the case of $\cdot OO\text{-}CH_2\text{-}O\text{-}SO_3^-$ where the tetroxide could readily decompose to form the $\cdot O\text{-}CH_2\text{-}O\text{-}SO_3^-$ radical in the singlet state, the fragmentation of $O_3^-S\text{-}O\text{-}CH(COOH)\text{-}OOOO\text{-}CH(COOH)\text{-}O\text{-}SO_3^-$ to form the alkoxyl radical $HOOC\text{-}CH(O)\cdot\text{-}O\text{-}SO_3^-$ occurred on the triplet state instead. The triplet electronic state was shown from a recent study to be favorable to the decomposition of tetroxides to alkoxyl radicals from some atmospherically relevant peroxyl radicals (Salo et al., 2022).

Line 338: The atmospheric implication appears to be somewhat incomplete with regard to the reaction of glycolic acid sulfate with respect to the rate constants.

We have added the rate constant for the reaction of glycolic acid sulfate in the revised manuscript.

Line 351

In the aqueous-phase, we determined rate constants of $7.87 \times 10^6$ M$^{-1}$s$^{-1}$ and $7.29 \times 10^8$ M$^{-1}$s$^{-1}$ for the reaction of methyl sulfate and glycolic acid sulfate, respectively.

Line 362: What do the authors mean with low reactivity of glycolic acid sulfate relative to methyl sulfate? The rate constant of glycolic acids with OH radicals was measured with k = 5x10^8 L mol-1 s-1 (Buxton et al., 1988). The rate constant of methyl sulfate with OH is in the range of 5x10^7 to 1x10^7 L mol-1s-1 (Gweme and Styler, 2024 (doi: 10.1021/acs.jpca.4c02877)). To what extent can the presence of the SO3 group deactivate the CH2 group of glycolic acid sulfate?

We have re-analyzed our findings based on previous results. As discussed in our reply to a previous comment of the Referee, we speculated that the difference between our reported rate constant for the HO• reaction with methyl sulfate may arise both from computational errors and environmental factors such as pH, temperature and ionic

strength. It is generally accepted that DFT calculations can induce up to 2-3 kcal mol$^{-1}$ errors in reaction energies can arise from DFT calculations (Bogojeski et al., 2020), which can induce up to two order of magnitude increase or decrease in reaction rate constants.

For glycolic acid, we determined a rate constant of $7.29 \times 10^8$ M$^{-1}$s$^{-1}$ in good agreement with the value measured by Buxton et al. (Buxton et al., 1988) and more than two orders of magnitude higher than the rate constant for the reaction of methyl sulfate. The following text is added to the revised manuscript to justify the difference between the rate constant from our calculations and the experimental data.

Line 234

This value is about 13 times lower than a previous experimental value (Gweme and Styler, 2024), and the difference can be attributed both to computational errors and environmental factors.

The sentence at line 304 is modified to include the bimolecular rate constant for the aqueous-phase reaction of glycolic acid sulfate.

A bimolecular rate constant of $7.29 \times 10^8$ M$^{-1}$s$^{-1}$ is determined for this reaction, in good agreement with the experimental report by Buxton et al. (Buxton et al., 1988).

The sentences at lines 362-365 are modified to:

Overall, the kinetics results show a moderate difference between the rate constants of HO• reactions with methyl sulfate and glycolic acid sulfate. The slightly high rate constant of the reaction of glycolic acid sulfate indicate the enhancing effect of the -COOH group in the hydrogen abstraction by HO•. Moreover, we found that the hydrogen abstraction from the -COOH group in glycolic acid sulfate leads to decarboxylation and eventually forms similar products as methyl sulfate. It can be inferred that for organosulfates with -COOH at β-position to the sulfate group, decarboxylation would be take place, leading to the formation of the corresponding alkyl sulfate radical. This highlights the potential role that chemical substitution on the carbon chain of organosulfates may play during their decomposition.

The sentence at lines 342-345 is updated to the following:

By investigating the decomposition mechanisms of two small atmospheric organosulfates (methyl sulfate and glycolic acid sulfate) by reaction with HO• radicals in this study, it was shown that the reaction of glycolic acid sulfate in the gas-phase is more kinetically favorable than that of methyl sulfate, which can be attributed to the effect of -COOH substitution that stabilizes the intermediate reactant complex from glycolic acid.

To summarise, taking all comments into account, I recommend the major revision of the manuscript. The topic itself is interesting and the manuscript has its merits, but in its current form is not sufficient.

We acknowledge all the valuable comments raised by the Referee. We have addressed all the comments, and we believe that our revised manuscript in its current state satisfactorily meets the standards of *Atmospheric Chemistry and Physics*.

**References**

Berndt, T., Scholz, W., Mentler, B., Fischer, L., Herrmann, H., Kulmala, M., and Hansel, A.: Accretion Product Formation from Self- and Cross-Reactions of $RO_2$ Radicals in the Atmosphere, 57, 3820-3824, 10.1002/anie.201710989, 2018.

Bogojeski, M., Vogt-Maranto, L., Tuckerman, M. E., Müller, K.-R., and Burke, K.: Quantum chemical accuracy from density functional approximations via machine learning, Nature Communications, 11, 5223, 10.1038/s41467-020-19093-1, 2020.

Bork, N., Kurtén, T., and Vehkamäki, H.: Exploring the atmospheric chemistry of $O_2SO_3^-$ and assessing the maximum turnover number of ion-catalysed $H_2SO_4$ formation, Atmospheric Chemistry and Physics, 13, 3695-3703, 2013.

Buehler, R. E., Staehelin, J., and Hoigne, J.: Ozone decomposition in water studied by pulse radiolysis. 1. Perhydroxyl ($HO_2$)/hyperoxide ($O_2^-$) and $HO_3/O_3^-$ as intermediates, The Journal of Physical Chemistry, 88, 2560-2564, 10.1021/j150656a026, 1984.

Buhler, R., Staehelin, J., and Hoigne, J.: Ozone Decomposition in Water Studied by Pulse Radiolysis 1. $HO_2/O_2^-$ and $HO_3/O_3^-$ as Intermediates - Correction, The Journal of Physical Chemistry, 88, 5450-5450, 10.1021/j150666a600, 1984.

Buxton, G. V., Greenstock, C. L., Helman, W. P., and Ross, A. B.: Critical Review of rate constants for reactions of hydrated electrons, hydrogen atoms and hydroxyl radicals ($\cdot OH/\cdot O^-$ in Aqueous Solution, Journal of Physical and Chemical Reference Data, 17, 513-886, 10.1063/1.555805, 1988.

Enghoff, M. and Svensmark, H.: The role of atmospheric ions in aerosol nucleation–a review, Atmospheric Chemistry and Physics, 8, 4911-4923, 2008.

England, C. and Corcoran, W. H.: Kinetics and Mechanisms of the Gas-Phase Reaction of Water Vapor and Nitrogen Dioxide, Industrial & Engineering Chemistry Fundamentals, 13, 373-384, 10.1021/i160052a014, 1974.

Fehsenfeld, F. and Ferguson, E.: Laboratory studies of negative ion reactions with atmospheric trace constituents, The Journal of Chemical Physics, 61, 3181-3193, 1974.

Ford, P. C. and Miranda, K. M.: The solution chemistry of nitric oxide and other reactive nitrogen species, Nitric Oxide, 103, 31-46, 10.1016/j.niox.2020.07.004, 2020.

Ghigo, G., Maranzana, A., and Tonachini, G.: Combustion and atmospheric oxidation of hydrocarbons: Theoretical study of the methyl peroxyl self-reaction, The Journal of Chemical Physics, 118, 10575-10583, 10.1063/1.1574316, 2003.

Goldman, M. J., Green, W. H., and Kroll, J. H.: Chemistry of Simple Organic Peroxy Radicals under Atmospheric through Combustion Conditions: Role of Temperature, Pressure, and NOx Level, The Journal of Physical Chemistry A, 125, 10303-10314, 10.1021/acs.jpca.1c07203, 2021.

Gweme, D. T. and Styler, S. A.: OH Radical Oxidation of Organosulfates in the

Atmospheric Aqueous Phase, The Journal of Physical Chemistry A, 128, 9462-9475, 10.1021/acs.jpca.4c02877, 2024.

Lai, D., Schaefer, T., Zhang, Y., Li, Y. J., Xing, S., Herrmann, H., and Chan, M. N.: Deactivating Effect of Hydroxyl Radicals Reactivity by Sulfate and Sulfite Functional Groups in Aqueous Phase—Atmospheric Implications for Small Organosulfur Compounds, ACS ES&T Air, 1, 678-689, 10.1021/acsestair.4c00033, 2024.

Lee, Y. N. and Schwartz, S. E.: Reaction kinetics of nitrogen dioxide with liquid water at low partial pressure, The Journal of Physical Chemistry, 85, 840-848, 10.1021/j150607a022, 1981.

Liang, Y.-N., Li, J., Wang, Q.-D., Wang, F., and Li, X.-Y.: Computational Study of the Reaction Mechanism of the Methylperoxy Self-Reaction, The Journal of Physical Chemistry A, 115, 13534-13541, 10.1021/jp2048508, 2011.

Lightfoot, P. D., Cox, R. A., Crowley, J. N., Destriau, M., Hayman, G. D., Jenkin, M. E., Moortgat, G. K., and Zabel, F.: Organic peroxy radicals: Kinetics, spectroscopy and tropospheric chemistry, Atmospheric Environment. Part A. General Topics, 26, 1805-1961, 10.1016/0960-1686(92)90423-I, 1992.

Lovato, M. E., Martín, C. A., and Cassano, A. E.: A reaction kinetic model for ozone decomposition in aqueous media valid for neutral and acidic pH, Chemical Engineering Journal, 146, 486-497, 10.1016/j.cej.2008.11.001, 2009.

Lv, G., Zhang, H., and Sun, X.: Insight into the formation of organosulfur compounds from the reaction of methyl vinyl ketone with sulfite radical in atmospheric aqueous phase, Science of The Total Environment, 774, 145817, 10.1016/j.scitotenv.2021.145817, 2021.

Lv, G., Yue, W., Wang, Z., Wang, G., Cheng, Z., Yang, Z., Xu, C., Qi, X., Cai, J., and Xu, X.: Mechanistic insight into the formation of aromatic organosulfates and sulfonates through sulfoxy-radical-initiated reactions in the atmospheric aqueous phase, Atmospheric Environment, 334, 120701, 10.1016/j.atmosenv.2024.120701, 2024.

Nangia, P. S. and Benson, S. W.: The kinetics of the interaction of peroxy radicals. II. Primary and secondary alkyl peroxy, International Journal of Chemical Kinetics, 12, 43-53, 10.1002/kin.550120105, 1980.

Russell, G. A.: Deuterium-isotope Effects in the Autoxidation of Aralkyl Hydrocarbons. Mechanism of the Interaction of PEroxy Radicals1, Journal of the American Chemical Society, 79, 3871-3877, 10.1021/ja01571a068, 1957.

Salo, V.-T., Valiev, R., Lehtola, S., and Kurtén, T.: Gas-Phase Peroxyl Radical Recombination Reactions: A Computational Study of Formation and Decomposition of Tetroxides, The Journal of Physical Chemistry A, 126, 4046-4056, 10.1021/acs.jpca.2c01321, 2022.

Seinfeld, J. H. and Pandis, S. N.: Atmospheric Chemistry and Physics: From Air Pollution to Climate Change, John Wiley & Sons, New York, 88 pp., 10.1063/1.882420, 1998.

Staehelin, J. and Hoigne, J.: Decomposition of ozone in water: rate of initiation by hydroxide ions and hydrogen peroxide, Environmental Science & Technology, 16, 676-681, 10.1021/es00104a009, 1982.

Staehelin, J., Buehler, R. E., and Hoigne, J.: Ozone decomposition in water studied by pulse radiolysis. 2. Hydroxyl and hydrogen tetroxide (HO4) as chain intermediates, The Journal of Physical Chemistry, 88, 5999-6004, 10.1021/j150668a051, 1984.

Svensmark, H., Pedersen, J. O. P., Marsh, N. D., Enghoff, M. B., and Uggerhoj, U. I.: Experimental evidence for the role of ions in particle nucleation under atmospheric conditions, Proceedings of the Royal Society a-Mathematical Physical and Engineering Sciences, 463, 385-396, 10.1098/rspa.2006.1773, 2007.

Tan, S. P. and Piri, M.: Modeling the Solubility of Nitrogen Dioxide in Water Using Perturbed-Chain Statistical Associating Fluid Theory, Industrial & Engineering Chemistry Research, 52, 16032-16043, 10.1021/ie402417p, 2013.

Tomaz, S., Wang, D., Zabalegui, N., Li, D., Lamkaddam, H., Bachmeier, F., Vogel, A., Monge, M. E., Perrier, S., Baltensperger, U., George, C., Rissanen, M., Ehn, M., El Haddad, I., and Riva, M.: Structures and reactivity of peroxy radicals and dimeric products revealed by online tandem mass spectrometry, Nature Communications, 12, 300, 10.1038/s41467-020-20532-2, 2021.

Tsona, N. T. and Du, L.: A potential source of atmospheric sulfate from $O_2^-$-induced $SO_2$ oxidation by ozone, Atmos. Chem. Phys., 19, 649-661, 10.5194/acp-19-649-2019, 2019.

Zhang, P., Wang, W., Zhang, T., Chen, L., Du, Y., Li, C., and Lü, J.: Theoretical Study on the Mechanism and Kinetics for the Self-Reaction of $C_2H_5O_2$ Radicals, The Journal of Physical Chemistry A, 116, 4610-4620, 10.1021/jp301308u, 2012.

**Reply to Anonymous Referee #3**

**Review of Tchinda et al for ACP**

Tchinda and co-workers study the transformation of glycolic acid sulfate and methyl sulfate both in the gas phase and particle phase using quantum chemical methods. The aqueous phase is modelled using a polarizable continuum model. Up to two explicit water molecules are considered in both the gas phase and aqueous phase calculations. The structure and vibrational frequencies of the stationary points are calculated at the M062X/6-311+g(2df,2pd) level of theory and the single point energies are refined using accurate CCSD(T)-F12/cc-pVDZ-F12 calculations.

This is an interesting topic that shed some further light on the atmospheric fate of organosulfates. However, the presentation is rather messy, and the study appears incomplete. Overall, I do not feel like I got wiser on the fate of organosulfates in the atmosphere and I cannot recommend publication in its current form.

**Major Comments**

Not all competing pathways are studied. It seems like only some selected pathways were considered. Why were the COOH hydrogen abstraction not considered? Why was the dimer formation only considered for one of the systems, but not the others, ect?

Initially, we explored all hydrogen-abstraction possibilities and while those from $CH_3$- and $-CH_2$- favorably formed alkyl radicals whose further chemistry can be assessed, the abstraction from -COOH led to $\bullet OC(O)-CH_2-O-SO_3H$ radical that further decomposes to methylene sulfate radical ($H_2C\bullet-O-SO_3H$) and $CO_2$. The chemistry of $H_2C\bullet-O-SO_3H$ radical was assessed in the section dealing with methyl sulfate. We observed that in the aqueous-phase, however, $\bullet OH$ acts as a bridge for hydrogen exchange between the carboxyl group and the sulfate group, making the hydrogen abstraction from -COOH not favorable in the aqueous-phase.

The following sentence is added in the Abstract, line 15
We found that the hydrogen abstraction from the -COOH group in glycolic acid sulfate could lead to decarboxylation and eventually form similar products as methyl sulfate.

The text at lines 295-300 have been modified to the following in the revised manuscript.
Hydrogen abstraction from glycolic acid sulfate could occur both from $-CH_2-$ and -COOH groups according to the following reactions:

$HOOC-CH_2-O-SO_3H + HO\bullet \rightarrow H_2O + HOOC-CH\bullet-O-SO_3H$ ,          (R9)

$HOOC-CH_2-O-SO_3H + HO\bullet \rightarrow H_2O + \bullet OC(O)-CH_2-O-SO_3H$ .          (R10)

The mechanism of reaction (R9) is similar to that of the hydrogen abstraction from methyl sulfate that forms an alkyl radical. Through this process, glycolic acid sulfate readily undergoes a hydrogen abstraction from the $-CH_2-$ group by HO•, resulting in $HOOC-CH\bullet-O-SO_3H$ formation. The reactant complex in this process lies at 4.65 kcal

mol$^{-1}$ at 298.15 K and 1 atm, and the transition state for its conversion is located 2.13 kcal mol$^{-1}$ above the reactant complex. We determined a bimolecular rate constant of the overall reaction is 6.17 ×10$^{-12}$ cm$^3$ molecule$^{-1}$ s$^{-1}$ for this reaction at 298.15 K. This shows that hydrogen abstraction by •OH from glycolic acid sulfate is more favorable than from methyl sulfate, hereby highlighting the enhancing effect of the carboxyl substituent. The further chemistry of HOOC-CH•-O-SO$_3$H is examined through reactions with O$_3$ and O$_2$. Our calculations show that contrary to the reaction with H$_2$C•-O-SO$_3$H, O$_3$ hardly reacts with HOOC-CH•-O-SO$_3$H as the O$_3$···HOOC-CH•-O-SO$_3$H formation is highly endergonic at standard conditions. However, the reaction with O$_2$ is seen to be fast, proceeding through formation of the reactant complex that is readily converted to HOOC-CH(OO)•-O-SO$_3$H. The observed negative Gibbs free energy barrier (-9.02 kcal mol$^{-1}$ below the reactant complex) in this conversion indicates that the formation of HOOC-CH(OO)•-O-SO$_3$H is almost instantaneous at standard conditions. Two molecules of HOOC-CH(OO)•-O-SO$_3$H develop into a tetroxide that then decomposes to HC(O)-O-SO$_3$H and HOOC-CH(O)•-O-SO$_3$H. The energetics of this reaction are provided in **Figure 6** and **Table S5**.

The abstraction from -COOH led to •OC(O)-CH$_2$-O-SO$_3$H (**reaction (R10)**) that further decomposes to methylene sulfate radical (H$_2$C•-O-SO$_3$H) and CO$_2$. The structures and energetics of all intermediate states of this reaction are given in **Figure S5** and **Table S6** in the Supplement. At the same level of theory, we determined a biomolecular rate constant of 3.86 ×10$^{-14}$ cm$^3$ molecule$^{-1}$ s$^{-1}$, two orders of magnitude lower than the hydrogen abstraction from the -CH$_2$- group. This indicates hydrogen that abstraction from glycolic acid would follow two competitive pathways although the hydrogen abstraction from the -CH$_2$- group preferred. It can be inferred that for organosulfates with -COOH at β-position to the sulfate group, decarboxylation would be a possible outcome of their decomposition initiated by •OH. The chemistry of H$_2$C•-O-SO$_3$H was assessed in Section 3.1 above.

We further investigated the HO•-initiated reaction of glycolic acid sulfate in the aqueous-phase, where the deprotonated state, HOOC-CH$_2$-O-SO$_3^-$, is predominant.

Line 362

Moreover, we found that the hydrogen abstraction from the -COOH group in glycolic acid sulfate leads to decarboxylation and eventually forms similar products as methyl sulfate. It can be inferred that for organosulfates with -COOH at β-position to the sulfate group, decarboxylation would be take place, leading to the formation of the corresponding alkyl sulfate radical.

Concerning the dimer formation from other systems, we previously focused on •OO-CH$_2$-O-SO$_3^-$ and HOOC-CH(OO•)-O-SO$_3^-$ (resulting from •OH-initiated oxidation of methyl sulfate and glycolic acid, respectively), exclusively, as we were unable to determine the transition states in the transformation of electrically neutral dimers, probably due to the effect of the complexed water. However, given that water is known to evaporate fast from gas-phase atmospheric species, we have re-considered the dimer formation without additional water and the results are now included in the revised

manuscript.

The following text is added at Line 222

Contrary to the self-decomposition of $H_2C\bullet$-O-SO$_3$H and reactions with NO$_2$ and O$_3$ that are favorable with $H_2C\bullet$-O-SO$_3$H hydrates, we were unable to fully optimize the O$_2$ reaction with $H_2C\bullet$-O-SO$_3$H hydrates but with the unhydrated system, instead. This led to exergonic formation of the $\bullet$OO-CH$_2$-O-SO$_3$H peroxyl radical with 45.98 kcal mol$^{-1}$ Gibbs free energy gain. The chemistry of peroxyl radicals has been the subject of several experimental and theoretical studies. It is widely accepted that the peroxyl radical would predominantly decompose to form alkoxyl radicals or alcohols along with carbonyl compounds through tetroxide intermediates (Russell, 1957). However, although the end products from this decomposition have been verified experimentally, the mechanisms have often been deemed unlikely due to inconsistency between thermodynamic experiments and computational studies (Nangia and Benson, 1980; Zhang et al., 2012; Liang et al., 2011). Moreover, the effort to elucidate alcohols and carbonyl compounds formation from the decomposition of the tetroxide in a previous study could not be achieved due to the impossibility to determine the corresponding transition states (Ghigo et al., 2003), while alkoxyl radical formation was observed to simply correspond to the dissociation of the tetroxide. A most recent theoretical study specifically focusing on the decomposition pathways of the tetroxide intermediate indicated that although substantial uncertainties may exist in their computed energetics, alkoxyl radicals are likely primary products from atmospherically relevant peroxyl radicals (Salo et al., 2022). Following the above reasoning, our calculations indicate that two molecules of $\bullet$OO-CH$_2$-O-SO$_3$H recombine to form a tetroxide that can further decompose to generate two alkoxyl radicals $\bullet$O-CH$_2$-O-SO$_3$H clustered to molecular oxygen. Then, the two $\bullet$O-CH$_2$-O-SO$_3$H radicals quickly interact with O$_2$ to form H$_2$O$_2$ and formic sulfuric anhydride (HC(O)-O-SO$_3$H) (**reaction (R5)**). The latter product has been identified to enhance new particle formation (An, 2024). Energetics and structures of all intermediates in this reaction are given in **Figure 3**.

Why are reactions with NO$_2$ and O$_3$ taken into account, but addition of O$_2$, which traditional tropospheric chemistry would dictate to be the most important, just neglected?

The fate of the radical ($H_2C\bullet$-O-SO$_3$H$\cdots$H$_2$O) resulting from $\bullet$OH reaction with the organosulfate was examined based on its estimated atmospheric lifetime relative to the self-decomposition process. As explained in the manuscript, the estimated atmospheric lifetimes based on our calculations indicated that $H_2C\bullet$-O-SO$_3$H would live enough to experience collisions with most atmospheric oxidants (including O$_2$, O$_3$ and NO$_2$) before self-decomposition could occur. However, despite previous attempts to investigate O$_2$ reaction with $H_2C\bullet$-O-SO$_3$H$\cdots$H$_2$O failed, this system has been re-considered without additional water molecule. Our calculations indicate that $H_2C\bullet$-O-SO$_3$H would react fast with O$_2$ to form a peroxy radical ($\bullet$OO-CH$_2$-O-SO$_3$H). Thereafter, two molecules of this radical would combine to form a tetroxide, which can further decompose to generate formic sulfuric anhydride (HOC-O-SO$_3$H). The following changes have been made in the revised manuscript to highlight the outcome of the O$_2$ + $H_2C\bullet$-O-SO$_3$H reaction.

The sentence at line 19 is modified to the following:
The primary reaction products are inorganic sulfate, carbonyl compounds and formic sulfuric anhydride.

The sentence at Line 21 is modified to the following:
Additionally, while prior studies suggested $O_2$ as primary oxidant in the fragmentation of organosulfates, this study unveils $O_3$ as a complementary oxidant in the intermediate steps of this process.

The following is added at line 150:
Considering that water is known to evaporate fast from atmospheric species, $H_2C\bullet\text{-O-SO}_3H$ would readily react both in its unhydrated and its hydrated forms.

The following reaction is added at Line 155:
$H_2C\bullet\text{-O-SO}_3H\cdots(H_2O)_{n+1} + O_2 \rightarrow \rightarrow HC(O)\text{-O-SO}_3H$ .               (R5)

The following text is added at Line 222
Contrary to the self-decomposition of $H_2C\bullet\text{-O-SO}_3H$ and reactions with $NO_2$ and $O_3$ that are favorable with $H_2C\bullet\text{-O-SO}_3H$ hydrates, we were unable to fully optimize the $O_2$ reaction with $H_2C\bullet\text{-O-SO}_3H$ hydrates but with the unhydrated system, instead. This led to exergonic formation of the $\bullet\text{OO-CH}_2\text{-O-SO}_3H$ peroxyl radical with 45.98 kcal $mol^{-1}$ Gibbs free energy gain. The chemistry of peroxyl radicals has been the subject of several experimental and theoretical studies. It is widely accepted that the peroxyl radical would predominantly decompose to form alkoxyl radicals or alcohols along with carbonyl compounds through tetroxide intermediates (Russell, 1957). However, although the end products from this decomposition have been verified experimentally, the mechanisms have often been deemed unlikely due to inconsistency between thermodynamic experiments and computational studies (Nangia and Benson, 1980; Zhang et al., 2012; Liang et al., 2011). Moreover, the effort to elucidate alcohols and carbonyl compounds formation from the decomposition of the tetroxide in a previous study could not be achieved due to the impossibility to determine the corresponding transition states (Ghigo et al., 2003), while alkoxyl radical formation was observed to simply correspond to the dissociation of the tetroxide. A most recent theoretical study specifically focusing on the decomposition pathways of the tetroxide intermediate indicated that although substantial uncertainties may exist in their computed energetics, alkoxyl radicals are likely primary products from atmospherically relevant peroxyl radicals (Salo et al., 2022). Following the above reasoning, our calculations indicate that two molecules of $\bullet\text{OO-CH}_2\text{-O-SO}_3H$ recombine to form a tetroxide that can further decompose to generate two alkoxyl radicals $\bullet\text{O-CH}_2\text{-O-SO}_3H$ clustered to molecular oxygen. Then, the two $\bullet\text{O-CH}_2\text{-O-SO}_3H$ radicals quickly interact with $O_2$ to form $H_2O_2$ and formic sulfuric anhydride ($HC(O)\text{-O-SO}_3H$) (**reaction (R5)**). The latter product has been identified to enhance new particle formation (An, 2024). Energetics and structures of all intermediates in this reaction are given in **Figure 3**.

Table 1 was revised to include the energetics of the reaction of the alkyl radical with $O_2$.

**Table 1: Electronic energy changes ($\Delta E$) and Gibbs free energy changes ($\Delta G$ at 298.15 K and 1 atm) for all intermediate species in the reaction of methyl sulfate**

with HO• radicals. Energy units are kcal mol$^{-1}$. "RC" stands for intermediate reactant complex, "TS" stands for transition state, "PD" stands for product, "nw" stands for the number of added water molecules to the reaction of methyl sulfate with HO• radicals.

| Reaction | $\Delta G$ | $\Delta E$ |
|---|---|---|
| CH$_3$-O-SO$_3$H···(H$_2$O)$_{n=0-2}$ + •OH ↔ RC1-nw → TS1-nw → PD1-nw (H$_2$C•-O-SO$_3$H···(H$_2$O)$_{n+1}$) | | |
| n = 0 | | |
| RC1-0w | 5.31 | -2.57 |
| TS1-0w | 11.76 | 5.66 |
| PD1-0w | -16.88 | -26.17 |
| n = 1 | | |
| RC1-1w | 6.23 | -13.38 |
| TS1-1w | 12.91 | -5.16 |
| PD1-1w | -12.01 | -32.34 |
| n = 2 | | |
| RC1-2w | 7.99 | -22.64 |
| TS1-2w | 11.53 | -18.10 |
| PD1-2w | -13.02 | -46.65 |
| | | |
| H$_2$C•-O-SO$_3^-$,H$_3$O$^+$···(H$_2$O)$_2$ → TS1b-2w → PD1b-2w (H$_2$CO···SO$_3$•$^-$,H$_3$O$^+$···(H$_2$O)$_2$) | | |
| H$_2$C•-O-SO$_3^-$,H$_3$O$^+$···(H$_2$O)$_2$ | 0 | 0 |
| TS1b-2w | 18.79 | 23.35 |
| PD1b-2w | 0.72 | 7.09 |
| | | |
| H$_2$C•-O-SO$_3$H + O$_3$ ↔ RC31→ TS31 → PD31 (O$_2$ + •O-CH$_2$-O-SO$_3$H) → RC31b → TS31b → PD31b | | |
| RC31 | -18.70 | -37.32 |
| TS31 | -25.72 | -42.17 |
| PD31 | -22.48 | -34.73 |
| RC31b | -4.48 | 5.22 |
| TS31b | 10.42 | 26.55 |

| | | |
|---|---|---|
| PD31b | -22.24 | -7.05 |

| | | |
|---|---|---|
| $H_2C \bullet$-O-$SO_3H$ + $O_2$ → $\bullet OO$-$CH_2$-O-$SO_3H$ | -45.98 | -65.69 |

| | | |
|---|---|---|
| $\bullet OO$-$CH_2$-O-$SO_3H$ + $\bullet OO$-$CH_2$-O-$SO_3H$ → RC32 → TS32 → PD32 | | |
| RC32 | -99.39 | 6.28 |
| TS32 | -83.49 | 15.33 |
| PD32 | -234.34 | -10.68 |

The sentence at line 205 is updated to:

Optimized structures and energetics of all reactions states in these reactions are given in **Figure 3**, **Figure S2**, **Table 1** and **Table S4**.

[Figure]

**Figure 3: Gibbs free energy changes (in kcal mol⁻¹) and optimized structures for all intermediates in the reaction of H₂C•-O-SO₃H with O₃ (top) and O₂ (bottom). The sulfur atom is in yellow, the oxygen atom is in red, the carbon atom is in grey and the hydrogen atom is in white.**

Gas-phase bimolecular kinetics are well-established, and the reactant complex normally does not enter the rate constant calculation. Hence, I find it curious that the authors treat the reactions as unimolecular and just start from the RC.

Section 2.2 of the manuscript provides details on how the reaction rate constants are calculated. Although in the manuscript we mostly report unimolecular rates constants (given by Eq. (3)) that describe how fast the reactant complex develops into the products, the bimolecular rate constant (given by Eq. (1)) fully describes the kinetics from the separate reactants to the product complex. The bimolecular rate constant takes into account both the formation of the reactant complex from separate reactants (expressed by the equilibrium constant) and the transformation of the reactant complex (expressed by the unimolecular rate constant).

I do not buy the presented argument that glycolic acid sulfate does not react with OH in the gas phase.

We thank the Referee for raising this. Our previous statement was based on the fact that our attempts to assess the outcome of HOOC-CH•-O-SO$_3$H···H$_2$O reaction with O$_3$/O$_2$ failed since we were neither able to determine the different transition states nor to obtain barrierless formation of products. This is probably due to the stabilizing effect of complexed water. Given that water is known to evaporate fast from gas-phase atmospheric species, we have re-considered the investigations of O$_3$/O$_2$ reactions with HOOC-CH•-O-SO$_3$H without additional water molecule. Please, refer to our reply to the Referee's first comment for our full response to this query.

What is the concentration of the organosulfates in the gas-phase compared to the condensed phase? Would they partition to the particle phase faster than they can react with OH radicals?

Despite we do not recall the report of specific organosulfates concentrations in the gas-phase, a study reported that out of measured organosulfates in Beijing, the contribution from the gas-phase was up to 11.6 % of the total organosulfates (Le Breton et al., 2018), suggesting that a significant fraction of organosulfates would always be retained in the gas-phase. This study showed that partitioning of organosulfates to the particle-phase depends on relative humidity and temperature. While increased relative humidity promotes partitioning into the particle-phase, high temperatures favor retention in the gas-phase. Another study showed that gas-to-particle partitioning is regulated by particle hygroscopicity (Ohno et al., 2022). It is then likely that a non-negligible fraction of organosulfates would react with OH in the gas-phase while another fraction would react in the particle-phase. We added the following in the revised manuscript.

Line 51 is revised to:
Organosulfates primarily exists in the particulate phase due to their low volatility (Estillore et al., 2016; George and Abbatt, 2010), although a non-negligible fraction has been shown to always be present in the gas-phase (Ehn et al., 2010; Le Breton et al., 2018) where they can react continuously with gas-phase oxidants (e.g., HO• radicals, O$_3$, and NO$_3$ radicals) at or near particle surfaces.

Line 65:
Both organosulfates have been detected at various locations around the world at concentrations in the ranges $1.08 \times 10^6$-$5.01 \times 10^7$ molecule cm$^{-3}$ for methyl sulfate (Hettiyadura et al., 2015; Peng et al., 2021) and $1.16 \times 10^7$-$4.71 \times 10^8$ molecule cm$^{-3}$ for glycolic acid sulfate (Huang et al., 2018; Hettiyadura et al., 2015; Wang et al., 2021; Hughes and Stone, 2019; Cai et al., 2020; Liao et al., 2015).

**Specific comments**

**Line 96**: In equation (3) quantum mechanical tunnelling is neglected. However, for

hydrogen abstraction reactions tunnelling would be expected to be important. Please include the role of tunnelling for the calculated rates.

We have included the tunneling correction in Eq. (3) and the following is added at line 100.

$$k_{\text{uni}} = \kappa \frac{k_B T}{h} \times \exp\left(-\frac{\Delta G^{\#}}{RT}\right). \tag{3}$$

$\kappa$ is the Eckart tunnelling coefficient (calculated by solving the Schrodinger equation for an asymmetrical one-dimensional Eckart potential (Eckart, 1930)).

**Line 104**: Equation (4)-(6) does not appear to be used in the manuscript.

This part of the manuscript provides details for calculating the bimolecular rate constants for aqueous-phase reactions. Numerical details for diffusion parameters are now provided in the revised Supplement and the following text is added in the revised manuscript at line 113:

All numerical values for radii, diffusion coefficients of reactants and the steady-state Smoluchowski rate constants are provided in **Tables S1** and **S2** in the Supplement.

**Line 134**: It would assume that the HO• + CH$_3$-O-SO$_3$H···(H$_2$O)n to H$_2$C•-O-SO$_3$H···(H$_2$O)n+1 reaction should be considered as a bimolecular reaction. Any particular reason why the reaction is treated as a unimolecular reaction starting from the reactant complex? Will this not misleading make it look like the reaction is more favourable than it is? Same analysis is made in line 211.

Indeed, CH$_3$-O-SO$_3$H···(H$_2$O)$_{n=0-2}$ reaction with HO• to H$_2$C•-O-SO$_3$H···(H$_2$O)$_{n+1}$ is bimolecular. Although in the manuscript we mostly report the unimolecular rate constant of the decomposition of the reactant complex that constitutes the limiting step, the overall hydrogen abstraction from the organosulfate is bimolecular and the rate constant for this process is given by Eq. (1). In the revised manuscript, we will make more reports of bimolecular rate constants when appropriate.

Line 139
The bimolecular rate constant for this reaction is determined to be $1.14 \times 10^{-13}$ cm$^3$ molecule$^{-1}$ s$^{-1}$ at 298.15 K.

Line 351
We also report a rate bimolecular rate constant of $6.17 \times 10^{-12}$ cm$^3$ molecule$^{-1}$ s$^{-1}$ for the gas-phase reaction of HO• reaction with glycolic acid at 298.15 K.

**Line 185**: Could you clarify what you mean with "… while attempts to optimize the reaction with O2 did not succeed due to electronic constraints to form chemically stable species." Are you referring to the spin states? Usually, this reaction occurs very fast and will therefore be more important than the reaction with O3.

The Referee is right that the reaction with O$_2$ should occur fast. In our previous

calculations, we treated the reaction by considering the direct product complex ($H_2C\bullet$-O-$SO_3H\cdots H_2O$) from the methyl sulfate reaction with HO$\bullet$. This reaction did not lead to a successful optimization of all the reaction states. Considering that water would evaporate fast from this product complex, we have performed a re-assessment of the fate of $H_2C\bullet$-O-$SO_3H$ in its unhydrated state and the $H_2C\bullet$-O-$SO_3H$ + $O_2$ reaction has been successfully investigated, along with the $H_2C\bullet$-O-$SO_3H$ + $O_3$ reaction.

We find that $O_2$ addition to $H_2C\bullet$-O-$SO_3H$ proceeds through formation of the peroxyl radical $\bullet OO$-$CH_2$-O-$SO_3H$ with 45.98 kcal mol$^{-1}$ Gibbs free energy gain. The fate of $\bullet OO$-$CH_2$-O-$SO_3H$ was further investigated.

The reaction with $O_3$ also proceeds through formation of the $O_3\cdots H_2C\bullet$-O-$SO_3H$ complex, following by an oxygen atom transfer to form the alkoxyl radical $\bullet O$-$CH_2$-O-$SO_3H$ and release $O_2$. $\bullet O$-$CH_2$-O-$SO_3H$ further undergoes intramolecular decomposition to form $H_2SO_4$ and HCO$\bullet$.

In the revised manuscript, the sentence at line 185 has been modified to the following:
Besides the self-decomposition of $H_2C\bullet$-O-$SO_3H$, we further examined its reactions with $NO_2$ (**reaction (R3)**), $O_3$ (**reaction (R4)**), and with $O_2$ (**reaction (R5)**).

The following text is added at Line 222
Contrary to the self-decomposition of $H_2C\bullet$-O-$SO_3H$ and reactions with $NO_2$ and $O_3$ that are favorable with $H_2C\bullet$-O-$SO_3H$ hydrates, we were unable to fully optimize the $O_2$ reaction with $H_2C\bullet$-O-$SO_3H$ hydrates but with the unhydrated system, instead. This led to exergonic formation of the $\bullet OO$-$CH_2$-O-$SO_3H$ peroxyl radical with 45.98 kcal mol$^{-1}$ Gibbs free energy gain. The chemistry of peroxyl radicals has been the subject of several experimental and theoretical studies. It is widely accepted that the peroxyl radical would predominantly decompose to form alkoxyl radicals or alcohols along with carbonyl compounds through tetroxide intermediates (Russell, 1957). However, although the end products from this decomposition have been verified experimentally, the mechanisms have often been deemed unlikely due to inconsistency between thermodynamic experiments and computational studies (Nangia and Benson, 1980; Zhang et al., 2012; Liang et al., 2011). Moreover, the effort to elucidate alcohols and carbonyl compounds formation from the decomposition of the tetroxide in a previous study could not be achieved due to the impossibility to determine the corresponding transition states (Ghigo et al., 2003), while alkoxyl radical formation was observed to simply correspond to the dissociation of the tetroxide. A most recent theoretical study specifically focusing on the decomposition pathways of the tetroxide intermediate indicated that although substantial uncertainties may exist in their computed energetics, alkoxyl radicals are likely primary products from atmospherically relevant peroxyl radicals (Salo et al., 2022). Following the above reasoning, our calculations indicate that two molecules of $\bullet OO$-$CH_2$-O-$SO_3H$ recombine to form a tetroxide that can further decompose to generate two alkoxyl radicals $\bullet O$-$CH_2$-O-$SO_3H$ clustered to molecular oxygen. Then, the two $\bullet O$-$CH_2$-O-$SO_3H$ radicals quickly interact with $O_2$ to form $H_2O_2$ and formic sulfuric anhydride (HC(O)-O-$SO_3H$) (**reaction (R5)**). The latter product has been identified to enhance new particle formation (An, 2024). Energetics and structures of all intermediates in this reaction are given in **Figure 3**.

**Line 231-232**: I assume the 1 M concentration is used to get the aqueous phase units right here. How well justified is it to use 1 M?

With quantum chemical calculations, while Gibbs free energies in the gas-phase are calculated at standard temperature of 298.15 K and standard pressure of 1 atm, in the aqueous-phase, they are obtained by converting the standard pressure of 1 atm to the standard concentration of 1 M.

This is updated in the revised manuscript.

Line 80.
In the aqueous-phase, the Gibbs free energies are calculated at standard temperature of 298.15 K and by converting the standard pressure of 1 atm (in the gas-phase) to the standard concentration of 1 M. Details are provided in Section S1 in the supplement.

**Line 280**: "its formation from $H_2C\bullet$-O-$SO_3^-$ reaction with $O_2$ is less thermodynamically favorable than with $O_3$,"

Our calculations indicated that the association of $H_2C\bullet$-O-$SO_3^-$ with $O_2$ (to form $\bullet OO$-$CH_2$-O-$SO_3^-$) is more thermodynamically favorable than the association with $O_3$ (to form $\bullet OOO$-$CH_2$-O-$SO_3^-$), with respective formation Gibbs free energies of -54.83 kcal mol$^{-1}$ and -52.05 kcal mol$^{-1}$. However, we find that $\bullet OOO$-$CH_2$-O-$SO_3^-$ can rapidly decompose to form the alkoxyl radical $\bullet O$-$CH_2$-O-$SO_3^-$, with a negative energy barrier of -5.98 kcal mol$^{-1}$, while $\bullet OO$-$CH_2$-O-$SO_3^-$ forms a tetroxide prior to $\bullet O$-$CH_2$-O-$SO_3^-$ formation. The 14.12 kcal mol$^{-1}$ Gibbs free energy separating the tetroxide from $\bullet O$-$CH_2$-O-$SO_3^-$ indicates a kinetically hindered process. This justifies our affirmation that $\bullet O$-$CH_2$-O-$SO_3^-$ formation from $H_2C\bullet$-O-$SO_3^-$ reaction with $O_2$ is less favorable than with $O_3$. In the revised manuscript, we change "thermodynamically" by "kinetically".

**Line 296**: "… HOOC-CH$\bullet$-O-$SO_3H$ species was found to be without atmospheric relevance as opposed to the intermediate reactant in the reaction of methyl sulfate."

I do not understand the argument here. The alkyl radical should be able to react barrierlessly with $O_2$?

We initially assessed the reaction of HOOC-CH$\bullet$-O-$SO_3H$ with $O_2$ by considering the direct product complex (HOOC-CH$\bullet$-O-$SO_3H\cdots H_2O$) of the glycolic acid reaction with HO$\bullet$. This reaction did not lead to a successful optimization of all the reaction stationary states. Given the known fast evaporation of water from most complexes in the gas-phase, we have re-considered the fate of HOOC-CH$\bullet$-O-$SO_3H$ in its unhydrated state and the HOOC-CH$\bullet$-O-$SO_3H$ + $O_2$ reaction has investigated.

The text at lines 296-301 has been deleted and the following new text is added in the revised manuscript.

Line 295

Hydrogen abstraction from glycolic acid sulfate could occur both from $-CH_2-$ and $-COOH$ groups according to the following reactions:

$$HOOC\text{-}CH_2\text{-}O\text{-}SO_3H + HO\bullet \rightarrow H_2O + HOOC\text{-}CH\bullet\text{-}O\text{-}SO_3H \text{ ,} \qquad (R9)$$

$$HOOC\text{-}CH_2\text{-}O\text{-}SO_3H + HO\bullet \rightarrow H_2O + \bullet OC(O)\text{-}CH_2\text{-}O\text{-}SO_3H \text{ .} \qquad (R10)$$

The mechanism of **reaction (R9)** is similar to that of the hydrogen abstraction from methyl sulfate that forms an alkyl radical. Through this process, glycolic acid sulfate readily undergoes a hydrogen abstraction from the $-CH_2-$ group by $HO\bullet$, resulting in $HOOC\text{-}CH\bullet\text{-}O\text{-}SO_3H$ formation. The reactant complex in this process lies at 4.65 kcal mol$^{-1}$ at 298.15 K and 1 atm, and the transition state for its conversion is located 2.13 kcal mol$^{-1}$ above the reactant complex. We determined a bimolecular rate constant of the overall reaction is $6.17\times10^{-12}$ cm$^3$ molecule$^{-1}$ s$^{-1}$ for this reaction at 298.15 K. This shows that hydrogen abstraction by $\bullet OH$ from glycolic acid sulfate is more favorable than from methyl sulfate, hereby highlighting the enhancing effect of the carboxyl substituent. The further chemistry of $HOOC\text{-}CH\bullet\text{-}O\text{-}SO_3H$ is examined through reactions with $O_3$ and $O_2$. Our calculations show that contrary to the reaction with $H_2C\bullet\text{-}O\text{-}SO_3H$, $O_3$ hardly reacts with $HOOC\text{-}CH\bullet\text{-}O\text{-}SO_3H$ as the $O_3\cdots HOOC\text{-}CH\bullet\text{-}O\text{-}SO_3H$ formation is highly endergonic at standard conditions. However, the reaction with $O_2$ is seen to be fast, proceeding through formation of the reactant complex that is readily converted to $HOOC\text{-}CH(OO)\bullet\text{-}O\text{-}SO_3H$. The observed negative Gibbs free energy barrier (-9.02 kcal mol$^{-1}$ below the reactant complex) in this conversion indicates that the formation of $HOOC\text{-}CH(OO)\bullet\text{-}O\text{-}SO_3H$ is almost instantaneous at standard conditions. Two molecules of $HOOC\text{-}CH(OO)\bullet\text{-}O\text{-}SO_3H$ develop into a tetroxide that then decomposes to $HC(O)\text{-}O\text{-}SO_3H$ and $HOOC\text{-}CH(O)\bullet\text{-}O\text{-}SO_3H$. The energetics of this reaction are provided in **Figure 6** and **Table S5**.

The abstraction from $-COOH$ led to $\bullet OC(O)\text{-}CH_2\text{-}O\text{-}SO_3H$ (reaction (R10)) that further decomposes to methylene sulfate radical ($H_2C\bullet\text{-}O\text{-}SO_3H$) and $CO_2$. The structures and energetics of all intermediate states of this reaction are given in **Figure S5** and **Table S6** in the Supplement. At the same level of theory, we determined a biomolecular rate constant of $3.86\times10^{-14}$ cm$^3$ molecule$^{-1}$ s$^{-1}$, two orders of magnitude lower than the hydrogen abstraction from the $-CH_2-$ group. This indicates that hydrogen abstraction from glycolic acid would follow two competitive pathways although the pathway leading to the alkyl radical is somewhat preferred. It can be inferred that for organosulfates that have a $-COOH$ substituent at the $\beta$-position relative to the sulfate group, decarboxylation would be a possible outcome of their decomposition. The chemistry of $H_2C\bullet\text{-}O\text{-}SO_3H$ was assessed in Section 3.1 above.

We further investigated the $HO\bullet$-initiated reaction of glycolic acid sulfate in the aqueous-phase, where the deprotonated state, $HOOC\text{-}CH_2\text{-}O\text{-}SO_3^-$, is predominant.

**Line 331**: Why is the dimer from 2 peroxy radicals only formed for glycolic acid sulfate and only in the aqueous phase? What are the alternative fates of peroxy radicals?

Kindly refer to our reply to the previous question for our reply to this question.

**Line 334**: "… does not fragment to form in the alkoxyl radical in the singlet state, but rather transits to the triplet state before fragmentation could occur, the energy difference between the two electronic states being as low as 7.60 kcal mol-1 ."

How is it determined that it "transits" to the triplet state here? Did you do the intersystem crossing calculations?

We wanted to express that from our calculations we could not get the fragmentation of $O_3^-S-O-CH(COOH)-OOOO-CH(COOH)-O-SO_3^-$ to happen in the singlet state but rather in the triplet state, the two states being separated by 7.60 kcal mol$^{-1}$ Gibbs free energy.
We have rephrased this sentence as follows:

However, we found that contrary to the case of $\cdot OO-CH_2-O-SO_3^-$ where the tetroxide could readily decompose to form the $\cdot O-CH_2-O-SO_3^-$ radical in the singlet state, the fragmentation of $O_3^-S-O-CH(COOH)-OOOO-CH(COOH)-O-SO_3^-$ to form the alkoxyl radical $HOOC-CH(O)\cdot-O-SO_3^-$ occurred on the triplet state instead. The triplet electronic state was shown from a recent study to be favorable to the decomposition of tetroxides to alkoxyl radicals from some atmospherically relevant peroxyl radicals (Salo et al., 2022).

**Technical comments**

"highlight" has been removed from the sentence.

Additionally, while prior studies suggested $O_2$ as primary oxidant in the fragmentation of organosulfates, this study unveils $O_3$ as a complementary oxidant in the intermediate steps of this process.

Line 44: Sentence starting with "Despite organosulfates have generally … " please rephrase.

The indicated sentence has been deleted and the sentence at line 51 is revised to the following to express the idea of line 44:

Organosulfates primarily exists in the particulate phase due to their low volatility (Estillore et al., 2016; George and Abbatt, 2010), although a non-negligible fraction has been shown to always be present in the gas-phase (Ehn et al., 2010; Le Breton et al., 2018) where they can react continuously with gas-phase oxidants (e.g., HO$\cdot$ radicals, $O_3$, and $NO_3$ radicals) at or near particle surfaces.

M062X/6-311+g(2df,2pd) has been changed to M06-2X/6-311+g(2df,2pd) in the whole revised manuscript.

Line 175: I find this part of the sentence hard to understand "… and the combination of Russell (Russell, 1957) and Bennett and Summers (Bennett and Summers, 1974) mechanisms we speculated by the authors to explain this formation.". Could you please rephrase.

This sentence has been rephrased and the revised form reads as follows:

The products predicted by our calculations were observed in a previous experimental study by Kwong et al. (Kwong et al., 2018) for the same reaction, and the mechanisms from Russell (Russell, 1957) and Bennett and Summers (Bennett and Summers, 1974) were speculated by the authors to explain this formation.

Line 188: "difficulty breaking" -> "difficulty in breaking"

This has been corrected.

Line 268: "oxygen atom of the alkoxy function" -> "oxygen atom of the alkoxy functional group"

This has been corrected.

**Data availability**

All coordinates of the studied systems as well as energetic should be available as supporting information. Otherwise, the study is not reproducible.

The cartesian coordinates of all studied systems are now added in the revised Supplement.

**References**

Bennett, J. E. and Summers, R.: Product Studies of the Mutual Termination Reactions of sec-Alkylperoxy Radicals: Evidence for Non-Cyclic Termination, Canadian Journal of Chemistry, 52, 1377-1379, 10.1139/v74-209, 1974.

Eckart, C.: The Penetration of a Potential Barrier by Electrons, Physical Review, 35, 1303-1309, 10.1103/PhysRev.35.1303, 1930.

Ehn, M., Junninen, H., Petaja, T., Kurten, T., Kerminen, V. M., Schobesberger, S., Manninen, H. E., Ortega, I. K., Vehkamaki, H., Kulmala, M., and Worsnop, D. R.: Composition and temporal behavior of ambient ions in the boreal forest, Atmospheric Chemistry and Physics, 10, 8513-8530, 10.5194/acp-10-8513-2010, 2010.

Estillore, A. D., Hettiyadura, A. P., Qin, Z., Leckrone, E., Wombacher, B., Humphry, T., Stone, E. A., and Grassian, V. H.: Water Uptake and Hygroscopic Growth of

Organosulfate Aerosol, Environ Sci Technol, 50, 4259-4268, 10.1021/acs.est.5b05014, 2016.

George, I. J. and Abbatt, J. P. D.: Heterogeneous oxidation of atmospheric aerosol particles by gas-phase radicals, Nature Chemistry, 2, 713-722, 10.1038/nchem.806, 2010.

Ghigo, G., Maranzana, A., and Tonachini, G.: Combustion and atmospheric oxidation of hydrocarbons: Theoretical study of the methyl peroxyl self-reaction, The Journal of Chemical Physics, 118, 10575-10583, 10.1063/1.1574316, 2003.

Kwong, K. C., Chim, M. M., Davies, J. F., Wilson, K. R., and Chan, M. N.: Importance of sulfate radical anion formation and chemistry in heterogeneous OH oxidation of sodium methyl sulfate, the smallest organosulfate, Atmos. Chem. Phys., 18, 2809-2820, 10.5194/acp-18-2809-2018, 2018.

Le Breton, M., Wang, Y., Hallquist, Å. M., Pathak, R. K., Zheng, J., Yang, Y., Shang, D., Glasius, M., Bannan, T. J., Liu, Q., Chan, C. K., Percival, C. J., Zhu, W., Lou, S., Topping, D., Wang, Y., Yu, J., Lu, K., Guo, S., Hu, M., and Hallquist, M.: Online gas- and particle-phase measurements of organosulfates, organosulfonates and nitrooxy organosulfates in Beijing utilizing a FIGAERO ToF-CIMS, Atmos. Chem. Phys., 18, 10355-10371, 10.5194/acp-18-10355-2018, 2018.

Liang, Y.-N., Li, J., Wang, Q.-D., Wang, F., and Li, X.-Y.: Computational Study of the Reaction Mechanism of the Methylperoxy Self-Reaction, The Journal of Physical Chemistry A, 115, 13534-13541, 10.1021/jp2048508, 2011.

Nangia, P. S. and Benson, S. W.: The kinetics of the interaction of peroxy radicals. II. Primary and secondary alkyl peroxy, International Journal of Chemical Kinetics, 12, 43-53, doi.org/10.1002/kin.550120105, 1980.

Ohno, P. E., Wang, J., Mahrt, F., Varelas, J. G., Aruffo, E., Ye, J., Qin, Y., Kiland, K. J., Bertram, A. K., Thomson, R. J., and Martin, S. T.: Gas-Particle Uptake and Hygroscopic Growth by Organosulfate Particles, ACS Earth and Space Chemistry, 6, 2481-2490, 10.1021/acsearthspacechem.2c00195, 2022.

Russell, G. A.: Deuterium-isotope Effects in the Autoxidation of Aralkyl Hydrocarbons. Mechanism of the Interaction of PEroxy Radicals1, Journal of the American Chemical Society, 79, 3871-3877, 10.1021/ja01571a068, 1957.

Salo, V.-T., Valiev, R., Lehtola, S., and Kurtén, T.: Gas-Phase Peroxyl Radical Recombination Reactions: A Computational Study of Formation and Decomposition of Tetroxides, The Journal of Physical Chemistry A, 126, 4046-4056, 10.1021/acs.jpca.2c01321, 2022.

Zhang, P., Wang, W., Zhang, T., Chen, L., Du, Y., Li, C., and Lü, J.: Theoretical Study on the Mechanism and Kinetics for the Self-Reaction of $C_2H_5O_2$ Radicals, The Journal of Physical Chemistry A, 116, 4610-4620, 10.1021/jp301308u, 2012.

**Author response to Editor's/Referee's comments on egusphere-2025-29**

While the quality of the manuscript has improved substantially, additional minor revisions have been requested to remove confusion:
Line 296: Could the authors clarify for the reader, what they meant by "environmental factors"? It is described in the answers to the reviewers, but a reader might not understand what was meant.

As discussed in our previous reply to the Referees' comments, rate constants determined experimentally could be effectively altered by environmental factors such as pH, temperature and ionic strength. This is now updated in the related sentence in the revised manuscript.

Page 11, line 270-272.

"This value is about 13 times lower than a previous experimental value (Gweme and Styler, 2024), and the difference can be attributed both to computational errors and environmental factors such as pH, temperature and ionic strength."

Line 300: The authors described well what is the fate of NO2 in aqueous solution and the solubility of O3, but how likely is this reaction pathway to play a role in terms of atmospheric concentrations as O2 is present?

Given the high atmospheric concentration of $O_2$ relative to those of $NO_2$ and $O_3$, the reaction with $O_2$ will definitely dominate the fate of $H_2C\bullet\text{-}O\text{-}SO_3^-$.
Regarding the reaction with $NO_2$, the following was added in the revised manuscript.

Page 12, line 279-281.
"Depending on environmental conditions, ozone can react via a direct reaction pathway involving molecular ozone or by an indirect route involving reactive intermediates that arise from its decomposition (Buehler et al., 1984; Buhler et al., 1984; Staehelin et al., 1984; Staehelin and Hoigne, 1982), whereas the reaction with $NO_2$ would be equivalent to explicitly assessing the reaction with $NO_3^-$."

Although the overall importance of the reaction with $O_2$ over the reaction with $O_3$ was already discussed in the previous version of the manuscript, it is further highlighted by modifying the following texts in the revised manuscript.

Page 1, line 21-23

"Additionally, besides $O_2$ as the primary oxidant in the fragmentation of organosulfates, this study unveils that $O_3$ may be a complementary oxidant in this process, especially in environments enriched with ozone."

Page 16, line 326-329:

"The alkoxy radical $\bullet O\text{-}CH_2\text{-}O\text{-}SO_3^-$ can readily decompose to form $HSO_4^-$ and $HCO\bullet$, by overcoming a barrier of 7.92 kcal mol$^{-1}$ and as explained above and exemplified in **Figure 5**, its formation from $H_2C\bullet\text{-}O\text{-}SO_3^-$ reaction with $O_2$ is less thermodynamically favorable than with $O_3$, despite the overall rate of the former should be higher than that of the latter due to the high atmospheric concentration of $O_2$."

Page 19, line 427-434:

"Among the three processes investigated (self-decomposition, reaction with $O_3$ and reaction with $O_2$), alkoxyl radicals can be formed both from alkyl radicals reactions with $O_2$ and with $O_3$. From the discussion above on reaction mechanisms and energetics, we clarify that beside $O_2$ reaction, the reaction with $O_3$ could be a complementary intermediate step in the formation of alkoxy radicals that further decompose to inorganic sulfate and carbonyl compounds. However, considering real atmospheric conditions (with $O_2$ concentrations much higher than $O_3$ concentrations), this formation will be kinetically driven by the reaction with $O_2$, whereas the reaction with $O_3$ may exclusively become relevant in environments highly enriched with ozone."